# STRUCTURING REPRESENTATION GEOMETRY WITH ROTATIONALLY EQUIVARIANT CONTRASTIVE LEARNING

**Sharut Gupta**[*]
MIT CSAIL
sharut@mit.edu

**Joshua Robinson**[*]
MIT CSAIL
joshrob@mit.edu

**Derek Lim**
MIT CSAIL
dereklim@mit.edu

**Soledad Villar**
Johns Hopkins University
svillar3@jhu.edu

**Stefanie jegelka**
MIT CSAIL
stefje@mit.edu

## ABSTRACT

Self-supervised learning converts raw perceptual data such as images to a compact space where simple Euclidean distances measure meaningful variations in data. In this paper, we extend this formulation by adding additional geometric structure to the embedding space by enforcing transformations of input space to correspond to simple (i.e., linear) transformations of embedding space. Specifically, in the contrastive learning setting, we introduce an *equivariance* objective and theoretically prove that its minima forces augmentations on input space to correspond to *rotations* on the spherical embedding space. We show that merely combining our equivariant loss with a non-collapse term results in non-trivial representations, without requiring invariance to data augmentations. Optimal performance is achieved by also encouraging approximate invariance, where input augmentations correspond to small rotations. Our method, CARE: **C**ontrastive **A**ugmentation-induced **R**otational **E**quivariance, leads to improved performance on downstream tasks, and ensures sensitivity in embedding space to important variations in data (e.g., color) that standard contrastive methods do not achieve. Code is available at https://github.com/Sharut/CARE.

## 1 INTRODUCTION

It is only partially understood what structure neural network representation spaces should possess in order to enable intelligent behavior to emerge efficiently (Ma et al., 2022). One known key ingredient is to learn low-dimensional spaces in which simple Euclidean distances effectively measure the similarity between data, as demonstrated by powerful self-supervised methods for web-scale learning (Chen et al., 2020; Schneider et al., 2021; Radford et al., 2021). However, many use cases require the use of richer structural relationships that similarities between data cannot capture. One example that has enjoyed considerable success is the encoding of relations between objects (*X is a parent of Y, A is a treatment for B*) as simple transformations of embeddings (e.g., translations), which has driven learning with knowledge graphs (Bordes et al., 2013; Sun et al., 2019; Yasunaga et al., 2022). But similar capabilities have been notably absent from existing self-supervised learning recipes.

Recent contrastive self-supervised learning approaches have explored ways to close this gap by ensuring representation spaces are sensitive to certain transformations of input data, such as variations in color (Dangovski et al., 2022; Devillers & Lefort, 2023; Garrido et al., 2023; Bhardwaj et al., 2023) that earlier contrastive methods lacked. Encouraging sensitivity is especially important in contrastive learning, as it is known to learn shortcuts that forget features that are not needed to solve the pretraining task (Robinson et al., 2021). This line of work formalizes sensitivity in terms of *equivariance*: transformations of input data correspond to predictable transformations in representation space. Equivariance requires specifying a family of transformations $a \in \mathcal{A}$ in the input space, a corresponding transformation $T_a$ in representation space, and training $f$ so that

---

[*]Equal contribution.

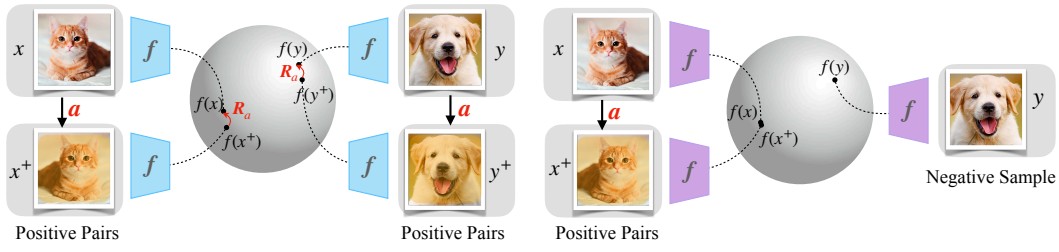

Figure 1: CARE is an equivariant contrastive learning approach that trains augmentations (cropping, blurring, etc.) of input data to correspond to orthogonal transformations of embedding space.

$f(a(x)) \approx T_a f(x)$. Prior works have considered a learnable feed-forward network for $T_a$, (Devillers & Lefort, 2023; Garrido et al., 2023). However, we find that this approach suffers from geometric pathologies, such as inconsistency under compositions: $T_{a_2 \circ a_1} f(x) \neq T_{a_2} T_{a_1} f(x)$. Furthermore, encoding the relation between the embeddings of $x$ and $a(x)$ in a non-linear way conflicts with the aim of learning a representation space where linear transformations relate different instances.

To address these concerns we propose CARE, an equivariant contrastive learning framework that learns to translate augmentations in the input space (such as cropping, blurring, and jittering) into simple *linear* transformations in feature space. Here, we use the sphere as our feature space (the standard contrastive learning space) and hence consider the isometries of the sphere: rotations and reflections, i.e., orthogonal transformations. As orthogonal transformations are less expressive (by design), our learning problem is more constrained, meaning that prior approaches for learning non-linear transforms cannot be used (see Section 3). Instead, CARE trains $f$ to preserve angles, i.e., $f(a(x))^\top f(a(x')) \approx f(x)^\top f(x')$, a property that must hold if $f$ is orthogonally equivariant. We show that achieving low error on this seemingly weaker property also implies approximate equivariance and enjoys consistency under compositions. Critically, we can easily integrate CARE into contrastive learning workflows since both operate by comparing pairs of data.

The key contributions of this work include:

- Introducing CARE, a novel equivariant contrastive learning framework that trains transformations (cropping, jittering, blurring, etc.) in input space to approximately correspond to local orthogonal transformations in representation space.

- Theoretically proving and empirically demonstrating that CARE places an orthogonally equivariant structure on the embedding space.

- Showing that CARE increases sensitivity to features (e.g., color) compared to invariance-based contrastive methods, and also improves performance on image recognition tasks.

## 2 RETHINKING HOW AUGMENTATIONS ARE USED IN SELF-SUPERVISED LEARNING

Given access only to samples from a marginal distribution $p(x)$ on some input space $\mathcal{X}$ such as images, the goal of representation learning is commonly to train a feature extracting model $f : \mathcal{X} \to \mathbb{S}^{d-1}$ mapping to the unit sphere $\mathbb{S}^{d-1} = \{z \in \mathbb{R}^d : \|z\|_2 = 1\}$. A common strategy to automatically generate supervision from the data is to additionally introduce a space of augmentations $\mathcal{A}$, containing maps $a : \mathcal{X} \to \mathcal{X}$ which slightly perturb inputs $\bar{x}$ (blurring, cropping, jittering, etc.). Siamese self-supervised methods learn representation spaces that reflect the relationship between the embeddings of $x = a(\bar{x})$ and $x^+ = a^+(\bar{x})$, commonly by training $f$ to be invariant or equivariant to the augmentations in the input space (Chen & He, 2021).

**Invariance to augmentation.** One approach is to train $f$ to embed $x$ and $x^+$ nearby—i.e., so that $f(x) = f(x^+)$ is *invariant* to augmentations. The InfoNCE loss (van den Oord et al., 2018; Gutmann

& Hyvärinen, 2010) used in contrastive learning achieves precisely this:

$$\mathcal{L}_{\text{InfoNCE}}(f) = \mathbb{E}_{x,x^+,\{x_i^-\}_{i=1}^N} \left[ -\log \frac{e^{f(x)^\top f(x^+)/\tau}}{e^{f(x)^\top f(x^+)/\tau} + \sum_{i=1}^N e^{f(x)^\top f(x_i^-)/\tau}} \right], \qquad (1)$$

where $\tau > 0$ is a temperature hyperparameter, and $x_i^- \sim p$ are negative samples from the marginal distribution on $\mathcal{X}$. As noted by Wang & Isola (2020), the contrastive training mechanism balances invariance to augmentations with a competing objective: uniformly distributing embeddings over the sphere, which rules out trivial solutions such as constant functions.

Whilst contrastive learning has produced considerable advances in large-scale learning (Radford et al., 2021), several lines of work have begun to probe the fundamental role of invariance in contrastive learning. Two key conclusions of recent investigations include: 1) invariance limits the expressive power of features learned by $f$, as it removes information about features or transformations that may be relevant in fine-grained tasks (Lee et al., 2021; Xie et al., 2022), and 2) contrastive learning actually benefits from not having exact invariance. For instance, a critical role of the projection head is to expand the feature space so that $f$ is not fully invariant (Jing et al., 2022), suggesting that it is preferable for the embeddings of $x$ and $x^+$ to be close, but not identical.

**Equivariance to augmentation.**  To address the limitations of invariance, recent work has additionally proposed to control *equivariance* (i.e., sensitivity) of $f$ to data transformations (Dangovski et al., 2022; Devillers & Lefort, 2023; Garrido et al., 2023). Prior works can broadly be viewed as training a set of features $f$ (sometimes alongside the usual invariant features) so that $f(a(x)) \approx T_a f(x)$ for samples $x \sim p$ from the data distribution where $T_a$ is some transformation of the embedding space. A common choice is to take $T_a f(x) = \text{MLP}(f(x), a)$, a learnable feed-forward network, and optimize a loss $\|\text{MLP}(f(x), a) - f(a(x))\|_2$. Whilst a learnable MLP ensures that information about $a$ is encoded into the embedding of $a(x)$, it permits complex non-linear relations between embeddings and hence does not necessarily encode relations in a linearly separable way. Furthermore, it does not enjoy the beneficial properties of equivariance in the formal group-theoretic sense, such as consistency under compositions in general: $T_{a_2 \circ a_1} f(x) \neq T_{a_2} T_{a_1} f(x)$.

Instead, this work introduces CARE, an equivariant contrastive learning approach respecting two key design principles:

**Principle 1.** *The map $T_a$ satisfying $f(a(x)) = T_a f(x)$ should be linear.*

**Principle 2.** *Equivariance should be learned from* pairs *of data, as in invariant contrastive learning.*

The first principle asks that $f$ converts complex perturbations $a$ of input data into much simpler (i.e., linear) transformations in embedding space. Specifically, we constrain the complexity of $T_a$ by considering isometries of the sphere, $O(d) = \{Q \in \mathbb{R}^{d \times d} : QQ^T = Q^T Q = I\}$, containing all rotations and reflections. Throughout this paper we define $f(a(x)) = T_a f(x)$ for $T_a \in O(d)$ to be *orthogonal equivariance*. This approach draws heavily from ideas in linear representation theory (Curtis & Reiner, 1966; Serre et al., 1977), which studies how to convert abstract group structures into matrix spaces equipped with standard matrix multiplication as the group operation.

The second principle stipulates *how* we want to learn orthogonal equivariance. Naively following previous non-linear approaches is challenging as our learning problem is more constrained, requiring learning a mapping $a \mapsto R_a$ to orthogonal matrices. Furthermore, for a single $(a, x)$ pair, the orthogonal matrix $R_a$ such that $f(a(x)) = R_a f(x)$ is not unique, making it hard to directly learn $R_a$. We sidestep these challenges by, instead of explicitly learning $R_a$, training $f$ so that an augmentation $a$ applied to two different inputs $x, x^+$ produces the same change in embedding space.

Our method, CARE, encodes data augmentations (cropping, blurring, jittering, etc.) as $O(d)$ transformations of embeddings using an equivariance-promoting objective function. CARE can be viewed as an instance of *symmetry regularization*, a term introduced by Shakerinava et al. (2022).

## 3  CARE: CONTRASTIVE AUGMENTATION-INDUCED ROTATIONAL EQUIVARIANCE

This section introduces a simple and practical approach for training a model $f : \mathcal{X} \to \mathbb{S}^{d-1}$ so that $f$ is orthogonally equivariant: i.e., a data augmentation $a \sim \mathcal{A}$ (cropping, blurring, jittering, etc.)

applied to any input $x \in \mathcal{X}$ causes the embedding $f(x)$ to be transformed by the same $R_a \in O(d)$ for all $x \in \mathcal{X}$: $f(a(x)) = R_a f(x)$.

To achieve this, we consider the following loss:

$$\mathcal{L}_{\text{equi}}(f) = \mathbb{E}_{a \sim \mathcal{A}} \mathbb{E}_{x,x' \sim \mathcal{X}} \left[ f(a(x'))^\top f(a(x)) - f(x)^\top f(x') \right]^2 \tag{2}$$

$$\mathcal{L}_{\text{equi}}(f) = \mathbb{E}_{a \sim \mathcal{A}} \mathbb{E}_{x,x' \sim \mathcal{X}} \left[ f(a(x'))^\top f(a(x)) - f(a'(x))^\top f(a'(x')) \right]^2 \tag{3}$$

Since inner products describe angles on the sphere, this objective enforces the angles between the embeddings of independent samples $x$ and $x'$ to be the same as those between their transformed counterparts $a(x)$ and $a(x')$. This is necessarily true if $f$ is orthogonally equivariant or, more generally, $R_a \in O(d)$ exists. But the converse—that $\mathcal{L}_{\text{equi}} = 0$ implies orthogonal equivariance—is non-obvious. In Section 3.1 we theoretically analyze $\mathcal{L}_{\text{equi}}$, demonstrating that it does indeed enforce mapping input augmentations to orthogonal transformations of embeddings. In practice, we replace the $f(x)^\top f(x')$ term with $f(a'(x))^\top f(a'(x'))$ for a freshly sampled $a' \sim \mathcal{A}$, noting that minimizing this variant also minimizes $\mathcal{L}_{\text{equi}}$, if we assume $a'$ can be the identity function with non-zero probability. A trivial but undesirable solution that minimizes $\mathcal{L}_{\text{equi}}$ is to collapse the embeddings of all points to be the same (see Figure 10). One natural approach to avoiding trivial solutions is to combine the equivariance loss with a non-collapse term such as the uniformity $\mathcal{L}_{\text{unif}}(f) = \log \mathbb{E}_{x,x' \sim \mathcal{X}} \exp\left( f(x)^\top f(x') \right)$ (Wang & Isola, 2020) whose optima $f$ distribute points uniformly over the sphere:

$$\mathcal{L}(f) = \mathcal{L}_{\text{equi}}(f) + \mathcal{L}_{\text{unif}}(f). \tag{4}$$

This is directly comparable to the InfoNCE loss, which can similarly be decomposed into two terms:

$$\mathcal{L}_{\text{InfoNCE}}(f) = \mathcal{L}_{\text{inv}}(f) + \mathcal{L}_{\text{unif}}(f) \tag{5}$$

where $\mathcal{L}_{\text{inv}}(f) = \mathbb{E}_{a,a' \sim \mathcal{A}} \| f(a(x)) - f(a'(x)) \|$ is minimized when $f$ is invariant to $\mathcal{A}$—i.e., $f(a(x)) = f(x)$. Figure 10 in the Appendix shows that training using $\mathcal{L}_{\text{equi}} + \mathcal{L}_{\text{unif}}$ yields non-trivial representations. However, the performance is below that of invariance-based contrastive learning approaches. We hypothesize that this is because data augmentations—which make small perceptual changes to data—should correspond to *small* perturbations of embeddings, which $\mathcal{L}_{\text{equi}}$ does not enforce.

To rule out this possibility, we introduce CARE: **C**ontrastive **A**ugmentation-induced **R**otational **E**quivariance. CARE additionally enforces the orthogonal transformations in embedding space to be *localized* by reintroducing an invariance loss term $\mathcal{L}_{\text{inv}}$ to encourage $f$ to be approximately invariant. Doing so breaks the indifference of $\mathcal{L}_{\text{equi}}$ between large and small rotations, biasing towards small. Specifically, we propose the following objective that combines our equivariant loss with InfoNCE:

$$\mathcal{L}_{\text{CARE}}(f) = \mathcal{L}_{\text{inv}}(f) + \mathcal{L}_{\text{unif}}(f) + \lambda \mathcal{L}_{\text{equi}}(f) \tag{6}$$

where $\lambda$ weights the equivariant loss. We note that many variations of this approach are possible. For instance, the equivariant loss and InfoNCE loss could use different augmentations, resulting in invariance to specific transformations while maintaining rotational equivariance to others, similar to Dangovski et al. (2022). The InfoNCE loss can also be replaced by other Siamese self-supervised losses. We leave further exploration of these possibilities to future work. In all, CARE consists of three components: (i) a term to induce orthogonal equivariance; (ii) a non-collapse term; and (iii) an invariance term to enforce localized transformations on the embedding space.

## 3.1 THEORETICAL PROPERTIES OF ORTHOGONALLY EQUIVARIANT LOSS

In this section, we establish that matching angles via $\mathcal{L}_{\text{equi}}$ leads to a seemingly stronger property. Specifically, $\mathcal{L}_{\text{equi}} = 0$ implies the existence of an orthogonal matrix $R_a \in O(d)$ for any augmentation $a$, such that $f(a(x)) = R_a f(x)$ holds for all $x$. The converse also holds and is easy to see. Indeed, suppose such an $R_a \in O(d)$ exists. Then, $f(a(x'))^\top f(a(x)) = f(x')^\top R_a^\top R_a f(x) = f(x)^\top f(x')$, which implies $\mathcal{L}_{\text{equi}}(f) = 0$. We formulate the first direction as a proposition.

**Proposition 1.** *Suppose $\mathcal{L}_{equi}(f) = 0$. Then for almost every $a \in \mathcal{A}$, there is an orthogonal matrix $R_a \in O(d)$ such that $f(a(x)) = R_a f(x)$ for almost all $x \in \mathcal{X}$.*

Figure 1 illustrates this result. Crucially $R_a$ is independent of $x$, without which the Proposition 1 would be trivial. That is, a single orthogonal transformation $R_a$ captures the impact of applying $a$ across the entire input space $\mathcal{X}$. Consequently, exact minimization of $\mathcal{L}_{\text{equi}}$ loss converts "unstructured" augmentations in input space to have a structured geometric interpretation as rotations in the embedding space. In Appendix B.1, we extend this result to the case where the loss is low but not exactly minimized, in which case we have a guarantee of approximate equivariance.

This result can be expressed as the existence of a mapping $\rho : \mathcal{A} \to O(d)$ that encodes the space of augmentations within $O(d)$. This raises a natural question: how much of the structure of $\mathcal{A}$ does this encoding preserve? For instance, assuming $\mathcal{A}$ is a semi-group (i.e., closed under compositions $a' \circ a \in \mathcal{A}$), does this transformation respect compositions: $f(a'(a(x))) = R_{a'} R_a f(x)$? This property does not hold for non-linear actions (Devillers & Lefort, 2023), but does for orthogonal equivariance:

**Corollary 1.** *If $\mathcal{L}_{equi}(f) = 0$ and $\{f(x) : x \in \mathcal{X}\}$ spans $\mathbb{R}^d$, then $\rho : \mathcal{A} \to O(d)$ given by $\rho(a) = R_a$ satisfies $\rho(a' \circ a) = \rho(a')\rho(a)$ for almost all $a, a'$. That is, $\rho$ defines a group action on $\mathbb{S}^{d-1}$ up to a set of measure zero.*

Formally, this result states that if $\mathcal{A}$ is a semi-group, then $\rho : \mathcal{A} \to O(d)$ defines a group homomorphism (or linear representation of $\mathcal{A}$ in the sense of representation theory (Curtis & Reiner, 1966; Serre et al., 1977), a branch of mathematics that studies the encoding of abstract groups as spaces of linear maps).

To exactly attain $\mathcal{L}_{\text{equi}}(f) = 0$, the space of augmentations $\mathcal{A}$ needs to have a certain structure, but this becomes less restrictive if $d$ is large. Assuming for simplicity that $\mathcal{A}$ is a group, the first isomorphism theorem for groups states that $\rho(\mathcal{A}) \simeq \mathcal{A}/\ker(\rho)$. For instance, if $\ker(\rho)$ is trivial, the equivariant loss can be exactly zero when the group of augmentations is a subgroup of the orthogonal group. Examples include orthogonal transformations or rotations that fix a subspace—i.e., $O(d')$ or $SO(d')$ with $d' \leq d$—or subgroups of the permutation group on $d$ elements. Furthermore, the Peter-Weyl theorem implies that any compact Lie group can be realized as a closed subgroup of $O(d)$ for some $d$ (Peter & Weyl, 1927). As with Proposition 1, we also extend this result to the case of low loss that is not exactly zero, where we show that compositionality is approximately preserved.

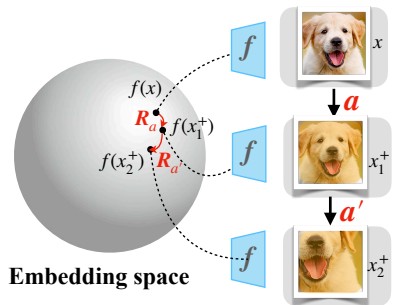

Figure 2: When $\mathcal{L}_{\text{equi}} = 0$, compositions of augmentations correspond to compositions of rotations.

## 3.2 EXTENSIONS TO OTHER GROUPS

Proposition 1 states that perfectly optimizing $\mathcal{L}_{\text{equi}} = 0$ produces an $f$ that is equivariant, encoding augmentations in the input space as orthogonal transformation in the embedding space. Notably, since the computation of $\mathcal{L}_{\text{equi}}$ solely relies on pairwise data instances $x, x' \in \mathcal{X}$, it naturally aligns with the contrastive learning paradigm that already works with pairs of data.

In fact, it is possible to extend the idea of CARE and its benefits to some other group actions. Mathematically, invariants of the action of $O(d)$ on $n$ points—seen in $(\mathbb{R}^d)^n$ as $Q(x_1, \dots x_n) = (Q x_1, \dots, Q x_n)$—can be expressed as a function of pairs of objects $(x_i^\top x_j)_{i,j=1\dots n}$. This is because the orthogonal group is defined as the stabilizer of a bilinear form. In other words, letting $B(x, x') = x^\top x'$ denote the standard inner product, we have

$$O(d) = \{A \in GL(d) : B(Ax, Ax') = B(x, x') \text{ for all } x, x' \in \mathbb{R}^d\}. \tag{7}$$

This argument applies more generally to other groups that are defined as stabilizers of bilinear forms. For instance, the Lorentz group, which has applications in the context of special relativity, can be defined as the stabilizer of the Minkowski inner product. Additionally, the symplectic group, which is used to characterize Hamiltonian dynamical systems, can be defined in a similar manner.

Such extensions to other groups allow to use CARE for different embedding space geometries. For instance, several recent works have used a hyperbolic space as an embedding space for self-supervised learners (Ge et al., 2022; Yue et al., 2023; Desai et al., 2023). If we constrain our embedding to a hyperboloid model of hyperbolic space, then linear isometries of this space are precisely the Lorentz group. Further discussions on extensions to other groups are given in Appendix D.

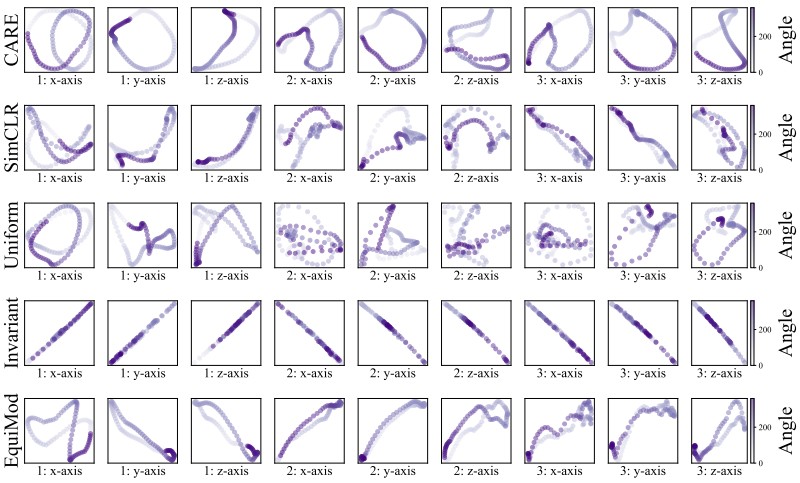

Figure 3: Trajectories through embedding space of three randomly sampled protein point clouds, rotated from $0$ to $2\pi$ in three orthogonal axes. Rows correspond to different training methods.

## 4 MEASURING ORTHOGONAL ACTION ON EMBEDDING SPACE

To probe the geometric properties of CARE, we consider two efficiently computable metrics for empirically measuring the orthogonal equivariance in the embedding space.

**Wahba's problem.** A natural way to assess the equivariance of $f$ is to sample a batch of data $\{x_i\}_{i=1}^n$ and an augmentation $a$ and test to what extent applying $a$ transforms the embeddings of each $x_i$ the same way. To measure this we compute a single rotation that approximates the map from $f(x_i)$ to $f(a(x_i))$ for all $i$. Let $F$ and $F_a \in \mathbb{R}^{d \times n}$ have $i$th columns $f(x_i)$ and $f(a(x_i))$ respectively, then we compute the error $\mathcal{W}_f = \min_{R \in SO(d)} \|RF - F_a\|_{\mathrm{Fro}}$, where $\|\cdot\|_{\mathrm{Fro}}$ denotes the Frobenius norm. If $\mathcal{W}_f = 0$, then $f(a(x_i)) = R_a f(x_i)$ for all $i$. This is a well-studied problem known as *Wahba's problem*. The analytic solution to Wahba's problem is computed easily. It is nearly $R^* = UV^\top$ where $U\Sigma V^\top$ is a singular value decomposition of $F_a F^\top$. However, a slight modification is required as this $R^*$ could have determinant $\pm 1$, and therefore may not belong to $SO(d)$. The only modification needed is to re-scale so that the determinant is one: $R^* = U \cdot \mathrm{diag}\{\mathbf{1}_{(n-1)}, \det(U)\det(V)\} \cdot V^\top$ where $\mathbf{1}_n$ denotes the vector in $\mathbb{R}^n$ of all ones.

**Relative rotational equivariance.** Optimizing for the CARE objective may potentially result in learning invariance since for input image $x$, $f(a(x)) = f(x)$ for $a \in \mathcal{A}$ is a trivial optimal solution of $\arg\min_f \mathcal{L}_{\mathrm{equi}}(f)$. To check that our model is learning non-trivial equivariance, we consider a metric similar to one proposed by Bhardwaj et al. (2023)

$$\gamma_f = \mathbb{E}_{a \sim \mathcal{A}} \mathbb{E}_{x,x' \sim \mathcal{X}} \left\{ \frac{(\|f(a(x')) - f(a(x))\|^2 - \|f(x') - f(x)\|^2)^2}{(\|f(a(x')) - f(x')\|^2 + \|f(a(x)) - f(x)\|^2)^2} \right\}. \tag{8}$$

Here, the denominator measures the invariance of the representation, with smaller values corresponding to greater invariance to the augmentations. The numerator, on the other hand, measures equivariance and can be simplified to $[f(a(x'))^\top f(a(x)) - f(x)^\top f(x')]^2$ (i.e., $\mathcal{L}_{\mathrm{equi}}(f)$) up to a constant, because $f$ maps to the unit sphere. The ratio $\gamma_f$ of these two terms measures the non-trivial equivariance, with a lower value implying greater non-trivial orthogonal equivariance.

## 5 EXPERIMENTS

We examine the representations learned by CARE, as well as those obtained from purely invariance-based contrastive approaches. We describe our experiment configurations in Appendix F.

## 5.1 LEARNING REPRESENTATIONS OF PROTEIN POINT CLOUDS

We consider the problem of learning representations of proteins from the Protein Data Bank (Burley et al., 2021). Each protein is described by a point cloud $X \in \mathbb{R}^{n \times 3}$. To respect the permutation invariance of each point cloud, we take $f$ to be a DeepSet (Zaheer et al., 2017), producing embeddings in $\mathbb{R}^{16}$, and train CARE using random rotations of 3D space as the augmentations—i.e., $X$ and $XR$ are a positive pair for $R \in SO(3)$. We evaluate our models on the task of predicting the first principal component of the point cloud, an important structural property of the input. This task is rotation equivariant, so we expect that CARE should outperform the invariance-based methods such as SimCLR, as is verified in Figure 4.

We test equivariance to rotations by randomly sampling a new protein $X$, and a sequence of rotations $\{R_i\}_{i=1}^{100}$ along each of the three orthogonal axes, evenly spaced, tracing a full $360°$ rotation of the point cloud. We then compute $z_i = f(XR_i)$ for each $i$, and project them into a 2D space. Each row of Figure 3 shows the trajectory of 3 different proteins, and three rotation trajectories, for a given training method. We find that CARE exhibits a much more regular geometry than models trained with SimCLR, $\mathcal{L}_{\text{unif}}$, or $\mathcal{L}_{\text{inv}}$. Learning the SO(3) manifold is challenging, and previous works assume access to the corresponding group action (Quessard et al., 2020; Park et al., 2021) However, CARE learns it by merely using $x$ and $a(x)$, without relying on the group action $a$.

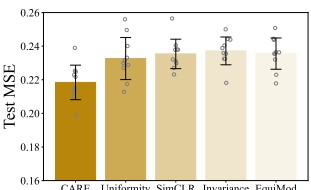

Figure 4: CARE achieves the lowest error on the task of predicting the first principal component of a protein

## 5.2 QUANTITATIVE MEASURES FOR ORTHOGONAL EQUIVARIANCE

**Wahba's Problem** We compare ResNet-18 models pretrained with CARE and with SimCLR on CIFAR10. For each model, we compute the optimal value $\mathcal{W}_f$ of Wahba's problem, as introduced in Section 4, over repeated trials. In each trial, we sample a single augmentation $a \sim \mathcal{A}$ at random and compute $\mathcal{W}_f$ for $f = f_{\text{CARE}}$ and $f = f_{\text{SimCLR}}$ over the test data. We repeat this process 20 times and plot the results in Figure 5, where the colors of dots indicate the sampled augmentation. Results show that CARE has a lower average error and worst-case error. Further, comparing point-wise for each augmentation, CARE achieves lower error in nearly all cases.

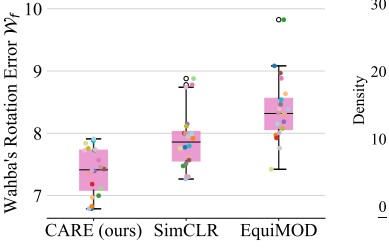
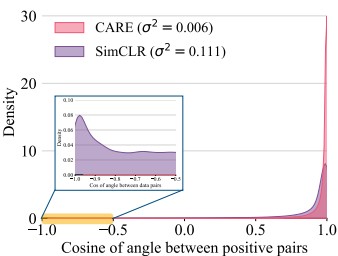
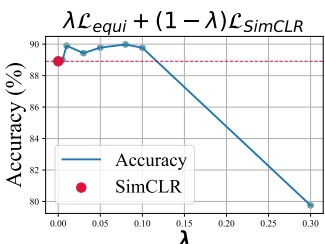

Figure 5: Measuring equivariance using Wahba's problem. Lower score is more equivariant.

Figure 6: Histogram of cosine angles between data pairs. CARE has much lower variance.

Figure 7: Linear readout error as the loss weightings vary.

**Analyzing structure on a 2D manifold.** To further study $\mathcal{L}_{\text{equi}}$, we train an encoder $f$ that projects the input onto a unit circle $\mathbb{S}^1$, where orthogonal transformations are defined by *angles*. We measure the cosine of the angle between pairs $f(x)$ and $f(a(x))$ for all $x$ in the test set, for 20 distinct sampled augmentations $a \sim \mathcal{A}$. As shown in Figure 6, Both CARE and SimCLR exhibit high density close to 1, demonstrating approximate invariance. However, unlike CARE, SimCLR exhibits non-zero density in the region $-0.5$ to $-1.0$, indicating that the application of augmentations significantly displaces the embeddings. Further, CARE consistently exhibits lower variance of the cosine between $f(x)$ and $f(a(x))$ for a fixed augmentation, showing that it transforms all embeddings in the same way.

**Ablation of loss terms.** The CARE loss $\mathcal{L}_{\text{CARE}}$ is a weighted sum of the InfoNCE loss $\mathcal{L}_{\text{InfoNCE}}$ and the orthogonal equivariance loss $\mathcal{L}_{\text{equi}}$. Figure 7 evaluates the performance of ResNet-50 models trained

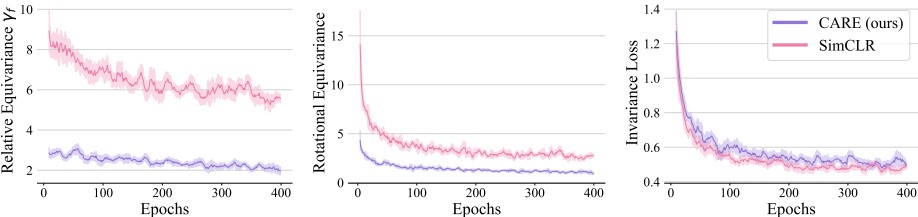

Figure 8: **Relative rotational equivariance** (Lower is more equivariant). Both CARE and invariance-based contrastive methods (e.g., SimCLR) produce *approximately* invariant embeddings. However, CARE learns a considerably more rotationally structured embedding space.

on CIFAR10 using $\mathcal{L}_{\text{InfoNCE}} + \lambda\mathcal{L}_{\text{equi}}$ for varying $\lambda$, finding optimal $\lambda$ in the range $0.01 \leq \lambda \leq 0.1$. We additionally split the InfoNCE loss into constituent parts $\mathcal{L}_{\text{inv}}$, $\mathcal{L}_{\text{unif}}$, and test different combinations of the three losses, including $\mathcal{L}_{\text{equi}}$. We find that using all three jointly is optimal. See Figure 10 in Appendix G.1 for detailed results.

**Relative rotational equivariance.** We measure the relative rotational equivariance for both CARE and SimCLR over the course of pretraining by following the approach outlined in Section 4. Specifically, we compare ResNet-18 models trained using CARE and SimCLR on CIFAR10. From Figure 8, we observe that both the models produce embeddings with comparable non-zero invariance loss $\mathcal{L}_{\text{inv}}$, indicating approximate invariance. However, they differ in their sensitivity to augmentations, with CARE attaining a much lower relative equivariance error $\gamma_f$. Importantly, this shows that CARE is *not* achieving lower equivariance error $\mathcal{L}_{\text{equi}}$ by collapsing to invariance, a trivial form of equivariance.

## 5.3 QUALITATIVE ASSESSMENT OF EQUIVARIANCE

A key property promised by equivariant contrastive models is sensitivity to specific augmentations. To qualitatively evaluate the sensitivity, or equivariance, of our models, we consider an image retrieval task on the Flowers-102 dataset (Nilsback & Zisserman, 2008), as considered by Bhardwaj et al. (2023). Specifically, when presented with an input image $x$, we extract the top 5 nearest neighbors based on the Euclidean distance of $f(x)$ and $f(a(x))$, where $a \in \mathcal{A}$. Figure 9 shows that retrieved results for the CARE model exhibit greater variability in response to a change in query color compared to the SimCLR model, which remains largely invariant. Additionally, Figure 21 compares CARE with EquiMOD (Devillers & Lefort, 2023), an equivariant baseline.

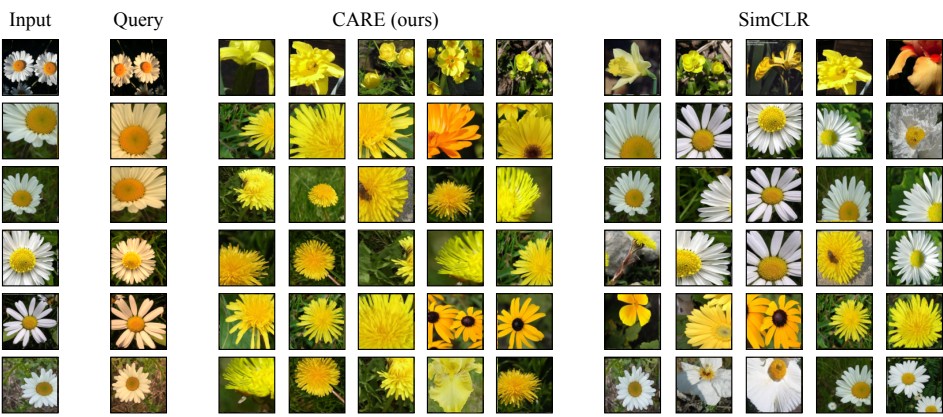

Figure 9: CARE exhibits sensitivity to features that invariance-based contrastive methods (e.g., SimCLR) do not. For each query input, we retrieve top 5 nearest neighbors in the embedding space.

## 5.4 LINEAR PROBE FOR IMAGE CLASSIFICATION

Next, we examine the quality of features learned by CARE for solving image classification tasks. We train ResNet-50 models on four datasets: CIFAR10, CIFAR100, STL10, and ImageNet100 using SimCLR and MoCo-v2 (see Appendix F for details). We refer to the model trained using CARE with SimCLR or MoCo-v2 backbone as CARE$_{SimCLR}$ and CARE$_{MoCo-v2}$ respectively. For each method and dataset, we evaluate the quality of the learned features by training a linear classifier (i.e., probe (Alain & Bengio, 2017)) on the frozen features of $f$ and report the performance on the test set in Table 1. We find consistent improvements in performance using CARE, showing the benefits of our structured embedding approach for image recognition tasks.

Table 1: Top-1 and Top-5 linear probe accuracy (%) on CIFAR10, CIFAR100, STL10 and ImageNet100 datasets. We report the mean performance from 3 different random initializations for the linear classifer. * denote numbers from Devillers & Lefort (2023), and ** from Zhuo et al. (2023)

| Method | CIFAR10 | CIFAR100 | STL10 | ImageNet100 |
|---|---|---|---|---|
| *Invariant prediction approaches* | | | | |
| SimCLR | $90.98_{\pm 0.10}$ | $66.77_{\pm 0.34}$ | $84.19_{\pm 0.13}$ | $72.79_{\pm 0.08}$ |
| MoCo-v2 | $91.95_{\pm 0.05}$ | $69.88_{\pm 0.23}$ | - | $73.50_{\pm 0.19}$ |
| BYOL | 90.44* | 67.41** | - | - |
| Barlow Twins | $84.54_{\pm 0.02}$ | $55.54_{\pm 0.05}$ | $90.62_{\pm 0.02}$ | |
| *Equivariant prediction approaches* | | | | |
| EquiMod$_{SimCLR}$ | 91.28 | 67.59 | 83.67 | - |
| EquiMod$_{BYOL}$ | 91.57* | - | - | - |
| CARE$_{SimCLR}$ | $\mathbf{91.92}_{\pm 0.12}$ ($\uparrow$ 0.94) | $\mathbf{68.05}_{\pm 0.28}$ ($\uparrow$ 1.28) | $\mathbf{84.64}_{\pm 0.29}$ ($\uparrow$ 0.45) | $\mathbf{76.69}_{\pm 0.08}$ ($\uparrow$ 3.90) |
| CARE$_{MoCo-v2}$ | $\mathbf{92.19}_{\pm 0.01}$ ($\uparrow$ 0.24) | $\mathbf{70.56}_{\pm 0.15}$ ($\uparrow$ 0.68) | $88.97_{\pm 0.48}$ | $\mathbf{74.30}_{\pm 0.07}$ ($\uparrow$ 0.80) |
| CARE$_{Barlow Twins}$ | $\mathbf{85.65}_{\pm 0.05}$ ($\uparrow$ 1.11) | $\mathbf{56.76}_{\pm 0.02}$ ($\uparrow$ 1.22) | $\mathbf{90.92}_{\pm 0.01}$ ($\uparrow$ 0.30) | - |

## 6 DISCUSSION

Converting transformations that are complex in input space into simple transformations in embedding space has many potential uses. For instance, modifying data (e.g., in order to reason about counterfactuals) can be viewed as transforming one embedding to another. If the sought after transformation was *simple* and *predictable*, it may be easier to find. Similarly, generalizing out-of-distribution is easier when extrapolating linearly (Xu et al., 2021), suggesting that linear transformations of embedding space may facilitate more reliable generalization. This work considers several design principles that may be broadly relevant: 1) *learned* equivariance preserves the expressivity of backbone architectures, and in some cases may be easier for model design than hard-coded equivariance, 2) linear group actions are desirable, but require carefully designed objectives (similar in spirit to the principle of *parsimony* (Ma et al., 2022), also advocated for by Shakerinava et al. (2022)), and 3) orthogonal (and related) symmetries are a promising structure for Siamese network training as they can be efficiently learned using *pair-wise* data comparisons.

## 7 ACKNOWLEDGEMENTS

This research was supported by NSF award CCF-2112665. Derek Lim is supported by National Science Foundation Graduate Research Fellowship. Soledad Villar is partially funded by the NSF–Simons Research Collaboration on the Mathematical and Scientific Foundations of Deep Learning (MoDL) (NSF DMS 2031985), NSF CISE 2212457, ONR N00014-22-1-2126 and an Amazon AI2AI Faculty Research Award.

We acknowledge MIT SuperCloud and Lincoln Laboratory Supercomputing Center for providing HPC resources that have contributed to this work. We wish to thank Michael Murphy for insightful discussions on extensions of our method to biology.

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

## A    RELATED WORK

**Geometry of representation space.** Equivariance is a key tool for encoding geometric structure—e.g., symmetries—into neural network representations (Cohen & Welling, 2016; Bronstein et al., 2021). Whilst hard-coding equivariance into model architectures is very successful, approximate learned equivariance (Kaba et al., 2022; Shakerinava et al., 2022), has certain advantages: 1) it applies when the symmetry is provided only by data, with no closed-form expression, 2) it can still be used when it is unclear how to hard code equivariance into the architecture, and 3) it can exploit standard high capacity architectures (He et al., 2016; Dosovitskiy et al., 2021), benefiting from considerable engineering efforts to optimize their performance. Shakerinava et al. (2022) also consider learning orthogonal equivariance but consider problems where both input and embedding space are acted on by $O(d)$. Our setting differs from this in two key ways: 1) we consider a very different set of transforms of input space—jitter, crops, etc.—and 2) CARE can be naturally integrated into contrastive learning, and 3) we theoretically study the minima of the angle-preserving loss. A related line of work, *mechanistic interpretability*, hypothesizes that algorithmic structure—possibly including group symmetries— emerges naturally within network connections during training (Chughtai et al., 2023). Our approach is very different from this as we directly *train* models to have the desired structure without relying on implicit processes.

**Self-supervised learning.** Prior equivariant contrastive learning approaches extend the usual setup of learning invariance by learning *sensitivity* to certain features known to be important for downstream tasks. For instance, Dangovski et al. (2022) learns to predict the augmentation applied but only considers a discrete group of 4-fold rotations. Lee et al. (2021) learns the difference of augmentation parameters and Xiao et al. (2021) constructs separate embedding sub-spaces that capture invariances to all but one augmentation. However, these approaches do not offer a meaningful structure to the embedding space. Others attempt to control how this sensitivity occurs. Specifically, Devillers & Lefort (2023); Garrido et al. (2023); Bhardwaj et al. (2023) learn a mapping from one latent representation to another, predicting how data augmentation affects the embedding. But this does not constrain the group action on embeddings, resulting in complex non-linear augmentation maps. Finally, the recent work Suau et al. (2023) implements approximate equivariance using 2D representations.

## B    PROOFS OF THEORETICAL RESULTS

The aim of this section is to detail the proofs of the theoretical results presented in the main manuscript. The key theoretical tools driving our analysis are prepared separately in Section C.

Throughout our analysis, we assume that all spaces (e.g., $\mathcal{A}$ and $\mathcal{X}$) are subspaces of Euclidean space and therefore admit a Lebesgue measure. We also assume that all distributions (e.g., $a \sim \mathcal{A}$ and $x \sim \mathcal{X}$) admit a density with respect to the Lebesgue measure. With these conditions in mind, we recall the loss function that is the main object of study:

$$\mathcal{L}_{\text{equi}}(f) = \mathbb{E}_{a \sim \mathcal{A}} \mathbb{E}_{x,x' \sim \mathcal{X}} \left[ f(a(x'))^\top f(a(x)) - f(x)^\top f(x') \right]^2 \tag{9}$$

Next, we re-state and prove Proposition 1, our first key result.

**Proposition 1.** *Suppose $\mathcal{L}_{equi}(f) = 0$. Then for almost every $a \in \mathcal{A}$, there is an orthogonal matrix $R_a \in O(d)$ such that $f(a(x)) = R_a f(x)$ for almost all $x \in \mathcal{X}$.*

*Proof.* Suppose that $\mathcal{L}_{\text{equi}}(f) = 0$. This means that $f(a(x'))^\top f(a(x)) = f(x)^\top f(x')$ for almost all $a \in G$, and $x, x' \in \mathcal{X}$. Setting $g_a(x) = f(a(x))$, we have that $g_a(x')^\top g_a(x) = f(x)^\top f(x')$. The continuous version of the First Fundamental Theorem of invariant theory for the orthogonal group (see Proposition 5) implies that there is an $R_a \in O(d)$ such that $f(a(x)) = g_a(x) = R_a f(x)$. □

As discussed in greater detail in the main manuscript, these results show that minimizing $\mathcal{L}_{\text{equi}}$ produces a model where an augmentation $a$ corresponds to a single orthogonal transformation of embeddings $R_a$, independent of the input. This result is continuous in flavor as it studies the loss over the full data distribution $p(x)$. There exists a corresponding result for the finite sample loss

$$\mathcal{L}_{\text{equi},n}(f) = \mathbb{E}_{a \sim \mathcal{A}} \sum_{i,j=1}^{n} \left[ f(a(x_j))^\top f(a(x_i)) - f(x_i)^\top f(x_j) \right]^2.$$

**Proposition 2.** *Suppose $\mathcal{L}_{equi,n}(f) = 0$. Then for almost every $a \in \mathcal{A}$, there is an orthogonal matrix $R_a \in O(d)$ such that $f(a(x_i)) = R_a f(x_i)$ for all $i = 1, \ldots, n$.*

As for the population counterpart, the proof of this result directly follows from the application of the First Fundamental Theorem of invariant theory for the orthogonal group.

*Proof of Proposition 2.* Suppose that $\mathcal{L}_{equi}(f) = 0$. This means that for almost every $a \in G$, and every $i, j = 1, \ldots, n$ we have $f(a(x_j))^\top f(a(x_i)) = f(x_i)^\top f(x_j)$. In other words $AA^T = BB^T$ where $A, B \in \mathbb{R}^{n \times d}$ are matrices whose $i$th rows are $A_i = f(a(x_i))^\top$ and $B_i = f(x_i)^\top$ respectively. This implies, by the First Fundamental Theorem of invariant theory for the orthogonal group (see Corollary 3), that there is an $R_a \in O(d)$ such that $A = BR_a$. Considering only the $i$th rows of $A$ and $B$ leads us to conclude that $f(a(x_i)) = R_a f(x_i)$. □

A corollary of Proposition 1 is that compositions of augmentations correspond to compositions of rotations.

**Corollary 1.** *If $\mathcal{L}_{equi}(f) = 0$ and $\{f(x) : x \in \mathcal{X}\}$ spans $\mathbb{R}^d$, then $\rho : \mathcal{A} \to O(d)$ given by $\rho(a) = R_a$ satisfies $\rho(a' \circ a) = \rho(a')\rho(a)$ for almost all $a, a'$. That is, $\rho$ defines a group action on $\mathbb{S}^{d-1}$ up to a set of measure zero.*

*Proof.* Applying Proposition 1 on $a' \circ a$ as the sampled augmentation, we have that $f(a' \circ a(x_i)) = R_{a' \circ a} f(x_i) = \rho(a' \circ a) f(x_i)$. However, taking $\bar{x} = a(x_i)$ and applying Proposition 1 twice we also know that $f(a' \circ a(x_i)) = f(a'(\bar{x})) = R_{a'} f(\bar{x}) = R_{a'} f(a(x_i)) = R_{a'} R_a f(x) = \rho(a')\rho(a) f(x_i)$. That is, $\rho(a' \circ a) f(x_i) = f(a' \circ a(x_i)) = \rho(a')\rho(a) f(x_i)$. Since this holds for all $i$ and the set of $f(x_i)$ spans $\mathbb{R}^d$, we have that $\rho(a' \circ a) = \rho(a')\rho(a)$. □

This corollary requires us to assume that $\mathcal{A}$ is a semi-group. That is, $\mathcal{A}$ is closed under compositions, but group elements do not necessarily have inverses and it does not need to include an identity element.

### B.1 Approximate Equivariance for Nonzero Loss

For Proposition 1, we show that when our equivariance loss is zero, the learned function is exactly equivariant. In practice, the equivariance loss may not exactly be minimized. Here, we give an approximate equivariance guarantee when the loss is small. The proof is based on Theorem 1 of Arias-Castro et al. (2020) (see Section C.1). For that, define

$$\mathcal{L}_{equi}(f, a) = \mathbb{E}_{x, x' \sim \mathcal{X}} \left[ f(a(x'))^\top f(a(x)) - f(x)^\top f(x') \right]^2 \tag{10}$$

**Proposition 3.** *Suppose we have a finite training set $\mathcal{X} = \{x_1, \ldots, x_n\}$. For an augmentation $a \in \mathcal{A}$, suppose that*

$$\mathcal{L}_{equi,\, n}(f, a) = \sum_{i=1}^n \sum_{j=1}^n \left[ f(a(x_i))^\top f(a(x_j)) - f(x_i)^\top f(x_j) \right]^2 \le \epsilon^4, \tag{11}$$

*and that the matrix $A = [f(x_1), \ldots, f(x_n)]^\top$ satisfies the conditions of Proposition 6. Let $A^\ddagger$ denote the pseudoinverse of $A$. Then there is an $R_a \in O(d)$ such that*

$$\sqrt{\sum_{i=1}^n \|f(a(x_i)) - R_a f(x_i)\|^2} \le (1 + \sqrt{2}) \|A^\ddagger\| \epsilon^2, \tag{12}$$

*and hence for each $x_i$,*

$$\|f(a(x_i)) - R_a f(x_i)\| \le (1 + \sqrt{2}) \|A^\ddagger\| \epsilon^2. \tag{13}$$

*Proof.* The first inequality is a direct application of Proposition 6, where $A = [f(x_1), \ldots, f(x_n)]^\top$ and $B = [f(a(x_1)), \ldots, f(a(x_n))]^\top$. The second inequality follows from the fact that $\|x\|_\infty \leq \|x\|_2$ for vectors $x \in \mathbb{R}^n$. $\qquad\square$

**Corollary 2.** *If $\mathcal{L}_{equi}(f, a) \leq \epsilon^4$ for all $a \in \mathcal{A}$ and $A = [f(x_1), \ldots, f(x_n)]^\top$ satisfies the conditions in Proposition 6 then $\rho : \mathcal{A} \to O(d)$ given by $\rho(a) = R_a$ satisfies $\|\rho(a' \circ a) - \rho(a')\rho(a)\| \leq o(\epsilon^2)$ for any $a, a' \in \mathcal{A}$.*

*Proof.* Applying Proposition 3 on $a' \circ a$ and using the triangle inequality, we have

$$
\begin{aligned}
\|R_{a' \circ a} f(x) - R_{a'} R_a f(x)\| &\leq \|R_{a' \circ a} f(x) - f(a' \circ a(x)) + f(a' \circ a(x)) - R_{a'} R_a f(x)\| \\
&\leq \|R_{a' \circ a} f(x) - f(a' \circ a(x))\| + \|f(a' \circ a(x)) - R_{a'} R_a f(x)\| \\
&\leq \|f(a' \circ a(x)) - R_{a'} R_a f(x)\| + (1 + \sqrt{2}) \|A^\ddagger\| \epsilon^2
\end{aligned}
$$

Now, using triangle inequality again, we have

$$
\begin{aligned}
\|R_{a' \circ a} f(x) - f(a' \circ a(x))\| &= \|f(a' \circ a(x)) - R_{a'} f(a(x)) + R_{a'} f(a(x)) - R_{a'} R_a f(x)\| \\
&\leq \|f(a' \circ a(x)) - R_{a'} f(a(x))\| + \|R_{a'} f(a(x)) - R_{a'} R_a f(x)\|
\end{aligned}
$$

Taking $\bar{x} = a(x)$, using the submultiplicativity property of matrix norms and applying Proposition 3 twice we have,

$$
\begin{aligned}
\|f(a' \circ a(x)) - R_{a'} f(a(x))\| &+ \|R_{a'} f(a(x)) - R_{a'} R_a f(x)\| \\
&= \|f(a'(\bar{x})) - R_{a'} f(\bar{x})\| + \|R_{a'} (f(a(x)) - R_a f(x))\| \\
&\leq \|f(a'(\bar{x})) - R_{a'} f(\bar{x})\| + \|R_{a'}\| \|(f(a(x)) - R_a f(x))\| \\
&\leq (1 + \sqrt{2}) \|\bar{A}^\ddagger\| \epsilon^2 + (1 + \sqrt{2}) \|A^\ddagger\| \epsilon^2
\end{aligned}
$$

Hence,

$$
\|R_{a' \circ a} f(x) - R_{a'} R_a f(x)\| \leq (1 + \sqrt{2}) \|\bar{A}^\ddagger\| \epsilon^2 + (2 + 2\sqrt{2}) \|A^\ddagger\| \epsilon^2
$$

where $A = [f(x_1), \ldots, f(x_n)]^\top$ and $\bar{A} = [f(a(x_1)), \ldots, f(a(x_n))]^\top$

By our assumption, we can choose a basis $f(x_{i_1}), f(x_{i_2}), \ldots f(x_{i_d})$ of $\mathbb{R}^d$, where $i_1, i_2, \ldots i_n \in [n]$. We can then write any unit norm $x$ in terms of this basis:

$$
x = \sum_{j=1}^{d} c_j f(x_{i_j}).
$$

Letting $B = [f(x_{i_1}), \ldots, f(x_{i_d})]^\top$, we have $B^{-1} x = c$. Thus, we can bound the 1-norm of $c$ as

$$
\|c\|_1 \leq \sqrt{d} \|c\|_2 \leq \sqrt{d} \|B^{-1}\|. \tag{14}
$$

Then using the triangle inequality, we have

$$
\begin{aligned}
\|R_{a' \circ a} x - R_{a'} R_a x\| &= \left\| \sum_{j=1}^{n} c_j \left( R_{a' \circ a} f(x_{i_j}) - R_{a'} R_a f(x_{i_j}) \right) \right\| \\
&\leq \sum_{j=1}^{n} |c_j| \|R_{a' \circ a} f(x_{i_j}) - R_{a'} R_a f(x_{i_j})\| \\
&\leq \|c\|_1 (1 + \sqrt{2}) \|A^\ddagger\| \epsilon^2 \\
&\leq \sqrt{d}(1 + \sqrt{2}) \|A^\ddagger\| \|B^{-1}\| \epsilon^2.
\end{aligned}
$$

Taking the supremum over all unit norm $x$ finishes the proof. $\qquad\square$

## C   BACKGROUND ON INVARIANCE THEORY FOR THE ORTHOGONAL GROUP

This section recalls some classical theory on orthogonal groups and an extension that we use for proving results over continuous data distributions.

A function $f : (\mathbb{R}^d)^n \to \mathbb{R}$ is said to be $O(d)$-invariant if $f(Rv_1, \ldots, Rv_n) = f(v_1, \ldots, v_n)$ for all $R \in O(d)$. Throughout this section, we are especially interested in determining easily computed statistics that *characterize* an $O(d)$ invariant function $f$. In other words, we would like to write $f$ as a function of these statistics. The following theorem was first proved by Hermann Weyl using Capelli's identity (Weyl, 1946) and shows that the inner products $v_i^\top v_j$ suffice.

**Theorem 4** (First fundamental theorem of invariant theory for the orthogonal group). *Suppose that* $f : (\mathbb{R}^d)^n \to \mathbb{R}$ *is* $O(d)$-*invariant. Then there exists a function* $g : \mathbb{R}^{n \times n} \to \mathbb{R}$ *for which*

$$f(v_1, \ldots, v_n) = g\big([v_i^\top v_j]_{i,j=1}^n\big).$$

In other words, to compute $f$ at a given input, it is not necessary to know all of $v_1, \ldots, v_n$. Computing the value of $f$ at a point can be done using only the inner products $v_i^\top v_j$, which are invariant to $O(d)$. Letting $V$ be the $n \times d$ matrix whose $i$th row is $v_i^\top$, we may also write $f(v_1, \ldots, n_n) = g(VV^\top)$. The map $V \mapsto VV^\top$ is known as the orthogonal projection of $V$.

A corollary of this result has recently been used to develop $O(d)$ equivariant architectures in machine learning (Villar et al., 2021).

**Corollary 3.** *Suppose that* $A, B$ *are* $n \times d$ *matrices and* $AA^\top = BB^\top$. *Then* $A = BR$ *for some* $R \in O(d)$.

Villar et al. (2021) use this characterization of orthogonally equivariant functions to *parameterize* function classes of neural networks that have the same equivariance. This result is also useful in our context; However, we put it to use for a very different purpose: studying $\mathcal{L}_{\text{equi}}$.

Intuitively this result says the following: given two point clouds $A, B$ of unit length vectors with some fixed correspondence (bijection) between each point in $A$ and a point in $B$, if the *angles* between the $i$th and $j$th points in cloud $A$ always equal the angle between the $i$th and $j$th point in cloud $B$, then $A$ and $B$ are the same up to an orthogonal transformation.

This is the main tool we use to prove the finite sample version of the main result for our equivariant loss (Proposition 2). However, to analyze the population sample loss $\mathcal{L}_{\text{equi}}$ (Proposition 1), we require an extended version of this result to the continuous limit as $n \to \infty$. To this end, we develop a simple but novel extension to Theorem 4 to the case of continuous data distributions. This result may be useful in other contexts independent of our setting.

**Proposition 5.** *Let* $\mathcal{X}$ *be any set and* $f, h : \mathcal{X} \to \mathbb{R}^d$ *be functions on* $\mathcal{X}$. *If* $f(x)^\top f(y) = h(x)^\top h(y)$ *for all* $x, y \in \mathcal{X}$, *then there exists* $R \in O(d)$ *such that* $Rf(x) = h(x)$ *for all* $x \in \mathcal{X}$.

The proof of this result directly builds on the finite sample version. The key idea of the proof is that since the embedding space $\mathbb{R}^d$ is finite-dimensional we may select a set of points $\{f(x_i)\}_i$ whose span has maximal rank in the linear space spanned by the outputs of $f$. This means that any arbitrary point $f(x)$ can be written as a linear combination of the $f(x_i)$. This observation allows us to apply the finite sample result on each $f(x_i)$ term in the sum to conclude that $f(x)$ is also a rotation of a sum of $h(x_i)$ terms. Next, we give the formal proof.

*Proof of Proposition 5.* Choose $x_1, \ldots, x_n \in \mathcal{X}$ such that $F = [f(x_1) \mid \ldots \mid f(x_n)]^\top \in \mathbb{R}^{n \times d}$ and $h = [h(x_1) \mid \ldots \mid h(x_n)]^\top \in \mathbb{R}^{n \times d}$ have maximal rank. Note we use "$\mid$" to denote the column-wise concatenation of vectors. Note that such $x_i$ can always be chosen. Since we have $FF^\top = HH^\top$, we know by Corollary 3 that $F = HR$ for some $R \in O(d)$.

Now consider an arbitrary $x \in \mathcal{X}$ and define $\tilde{F} = [F \mid f(x)]^\top$ and $\tilde{H} = [H \mid h(x)]^\top$, both of which belong to $\mathbb{R}^{(n+1) \times d}$. Note that again we have $\tilde{F}\tilde{F}^\top = \tilde{H}\tilde{H}^\top$ so also know that $\tilde{F} = \tilde{H}\tilde{R}$ for some $\tilde{R} \in O(d)$. Since $x_i$ were chosen so that $F$ and $H$ are of maximal rank, we know that $h(x) = \sum_{i=1}^n c_i h(x_i)$ for some coefficients $c_i \in \mathbb{R}$, since if this were not the case then we would have $\text{rank}(\tilde{H}) = \text{rank}(H) + 1$.

From this, we know that

$$
\begin{aligned}
R^\top h(x) &= \sum_{i=1}^{n} c_i R^\top h(x_i) \\
&= \sum_{i=1}^{n} c_i f(x_i) \\
&= \sum_{i=1}^{n} c_i \tilde{R}^\top h(x_i) \\
&= \tilde{R}^\top \sum_{i=1}^{n} c_i h(x_i) \\
&= \tilde{R}^\top h(x) \\
&= f(x).
\end{aligned}
$$

So we have that $Rf(x) = RR^\top h(x) = h(x)$ for all $x \in \mathcal{X}$. $\qquad\square$

## C.1 Approximate Equivariance Results

Here, we consider the setting when our equivariance loss is not exactly minimized. This corresponds to when the pairwise dot products between representations do not exactly match. We use a generalization of Corollary 3 to this case (Arias-Castro et al., 2020) Instead of the dot products $AA^\top$ and $BB^\top$ exactly matching, they match only up to some error. As a result, we can guarantee $A$ is close to $BR$ for some orthogonal matrix $R \in O(d)$ up to some error.

**Proposition 6.** *(Arias-Castro et al., 2020) Let $A, B \in \mathbb{R}^{n \times d}$ with $n \geq d$, where $A$ is full rank. Suppose $\left\| AA^\top - BB^\top \right\|_F \leq \epsilon^2$ and $\left\| A^\ddagger \right\| \epsilon \leq \frac{1}{\sqrt{2}}$ Then there exists an orthogonal $R \in O(d)$ such that*

$$
\left\| A - BR \right\|_F \leq (1 + \sqrt{2}) \left\| A^\ddagger \right\| \epsilon^2, \tag{15}
$$

*where $A^\ddagger$ is the pseudoinverse of $A$, $\left\| \cdot \right\|_F$ is the Frobenius norm, and $\left\| \cdot \right\|$ is the maximum singular value norm.*

In other words, for a fixed and full rank matrix $A$, we have that $\min_{R \in O(d)} \left\| A - BR \right\|_F = o(\epsilon)$, where $o(\epsilon) \to 0$ as the error $\epsilon \to 0$.

## D Extensions to other groups: further discussion

In Section 3.2, we explore the possibility of formulating an equivariant loss $\mathcal{L}_{\text{equi}}$ for pairs of points that fully captures equivariance by requiring the group to be the stabilizer of a bilinear form. In this context, the invariants are generated by polynomials of degree two in two variables, and the equivariant functions can be obtained by computing gradients of these invariants (Blum-Smith & Villar, 2022). Section 3.2 notes that this holds true not only for the orthogonal group, which is the primary focus of our research but also for the Lorentz group and the symplectic group, suggesting natural extensions of our approach.

It is worth noting that the group of rotations $SO(d)$ does not fall into this framework. It can be defined as the set of transformations that preserve both inner products (a 2-form) and determinants (a $d$-form). Consequently, some of its generators have degree 2 while others have degree $d$ (see (Weyl, 1946), Section II.A.9).

Weyl's theorem states that if a group acts on $n$ copies of a vector space (in our case, $(\mathbb{R}^d)^n$ for consistency with the rest of the paper), its action can be characterized by examining how it acts on $k$ copies (i.e., $(\mathbb{R}^d)^k$) when the maximum degree of its irreducible components is $k$ (refer to Section 6 of (Schmid, 2006) for a precise statement of the theorem). Since our interest lies in understanding equivariance in terms of pairs of objects, we desire invariants that act on pairs of

points. One way to guarantee this is to restrict ourselves to groups that act through representations where the irreducible components have degrees of at most two (though this is not necessary in all cases, such as the orthogonal group $O(d)$ that we consider in the main paper). An example of such groups is the product of finite subgroups of the unitary group $U(2)$, which holds relevance in particle physics. According to Weyl's theorem, the corresponding invariants can be expressed as *polarizations* of degree-2 polynomials on two variables. Polarizations represent an algebraic construction that enables the expression of homogeneous polynomials in multiple variables by introducing additional variables to polynomials with fewer variables. In our case, the base polynomials consist of degree-2 polynomials in two variables, while the polarizations incorporate additional variables. Notably, an interesting open problem lies in leveraging this formulation for contrastive learning.

## E    IMPLEMENTATION DETAILS

Algorithm 1 presents pytorch-based pseudocode for implementing CARE. This implementation introduces the idea of using a smaller batch size for the equivariance loss compared to the InfoNCE loss. Specifically, by definition, the equivariance loss is defined as a double expectation, one over data pairs and the other over augmentations. Empirical observations reveal that sampling one augmentation per batch leads to unstable yet superior performance when compared to standard invariant-based baselines such as SimCLR. Since these invariant-based contrastive benchmarks generally perform well with large batch sizes, we adopt the approach of splitting a batch into multiple chunks to efficiently sample multiple augmentations per batch for the equivariance loss. Each chunk of the batch is associated with a new pair of augmentations, ensuring a large batch size for the InfoNCE loss and a smaller batch size for the equivariance loss.

---

**Algorithm 1** PyTorch based pseudocode for CARE

1: **Notations:** $f$ represents the backbone encoder network, $\lambda$ is the weight on CARE loss, `apply_same_aug` function applies the same augmentation to all samples in the input batch
2: **for** minibatch $x$ in `dataloader` **do**
3:     draw *two batches* of augmentation functions $a_1, a_2 \in \mathcal{A}$
4:     /* Functions $a_1, a_2$ apply different augmentation to each sample in batch $x$ */
5:     $z_1^{\text{inv}}, z_2^{\text{inv}} = f(a_1(x)), f(a_2(x))$
6:     divide $x$ into `n_split` chunks to form $x_{\text{chunks}}$
7:     /* Module for calculating orthogonal equivariance loss */
8:     **for** $c_i$ in $x_{\text{chunks}}$ in parallel **do**
9:         draw *two* augmentation functions $\tilde{a}_1, \tilde{a}_2 \in \mathcal{A}$
10:         /* Functions $\tilde{a}_1, \tilde{a}_2$ apply same augmentation to each sample in batch $c_i$ */
11:         $\tilde{z}_{i1}, \tilde{z}_{i2} = f(\texttt{apply\_same\_aug}(c_i, \tilde{a}_1)), f(\texttt{apply\_same\_aug}(c_i, \tilde{a}_2))$
12:     /* Concatenate embedding vectors corresponding to all chunks */
13:     merge $\tilde{z}_{i1}, \tilde{z}_{i2}$ into $z_1^{\text{equiv}}, z_2^{\text{equiv}}$ respectively
14:     /* Loss computation */
15:     $\mathcal{L}_{\text{InfoNCE}}(f) = \texttt{infonce\_loss}(z_1^{\text{inv}}, z_2^{\text{inv}})$
16:     $\mathcal{L}_{\text{equiv}}(f) = \texttt{orthogonal\_equivariance\_loss}(z_1^{\text{equiv}}, z_2^{\text{equiv}}, \texttt{n\_split})$
17:     $\mathcal{L}_{\text{CARE}}(f) = \mathcal{L}_{\text{InfoNCE}}(f) + \lambda \cdot \mathcal{L}_{\text{equiv}}(f)$
18:     /* Optimization step */
19:     $\mathcal{L}_{\text{CARE}}(f)$.backward()
20:     optimizer.step()

---

## F    SUPPLEMENTARY EXPERIMENTAL DETAILS AND ASSETS DISCLOSURE

### F.1    ASSETS

We do not introduce new data in the course of this work. Instead, we use publicly available widely used image datasets for the purposes of benchmarking and comparison.

### F.2    HARDWARE AND SETUP

All experiments were performed on an HPC computing cluster using 4 NVIDIA Tesla V100 GPUs with 32GB accelerator RAM for a single training run. The CPUs used were Intel Xeon Gold 6248

processors with 40 cores and 384GB RAM. All experiments use the PyTorch deep learning framework (Paszke et al., 2019).

### F.3 EXPERIMENTAL PROTOCOLS

We first outline the training protocol adopted for training our proposed approach on a variety of datasets, namely CIFAR10, CIFAR100, STL10, and ImageNet100.

**CIFAR10, CIFAR100 and STL10**    All encoders have ResNet-50 backbones and are trained for 400 epochs with temperature $\tau = 0.5$ for SimCLR and $\tau = 0.1$ for MoCo-v2 [*]. The encoded features have a dimension of 2048 and are further processed by a two-layer MLP projection head, producing an output dimension of 128. A batch size of 256 was used for all datasets. For CIFAR10 and CIFAR100, we employed the Adam optimizer with a learning rate of $1e^{-3}$ and weight decay of $1e^{-6}$. For STL10, we employed the SGD optimizer with a learning rate of $0.06$, utilizing cosine annealing and a weight decay of $5e^{-4}$, with 10 warmup steps. We use the same set of augmentations as in SimCLR (Chen et al., 2020). To train the encoder using $\mathcal{L}_{\text{CARE-SimCLR}}$, we use the same hyper-parameters for InfoNCE loss. Additionally, we use 4, 8 and 16 batch splits for CIFAR100, STL10 and CIFAR10, respectively. This allows us to sample multiple augmentations per batch, effectively reducing the batch size of equivariance loss whilst retaining the same for InfoNCE loss. Furthermore, for the equivariant term, we find it optimal to use a weight of $\lambda = 0.01, 0.001,$ and $0.01$ for CIFAR10, CIFAR100, and STL10, respectively.

**ImageNet100**    We use ResNet-50 as the encoder architecture and pretrain the model for 200 epochs. A base learning rate of 0.8 is used in combination with cosine annealing scheduling and a batch size of 512. For MoCo-v2, we use 0.99 as the momentum and $\tau = 0.2$ as the temperature. All remaining hyperparameters were maintained at their respective official defaults as in the official MoCo-v2 code. While training with $\mathcal{L}_{\text{CARE-SimCLR}}$ and $\mathcal{L}_{\text{CARE-MoCo}}$, we find it optimal to use splits of 4 and 8 and weight of $\lambda = 0.005$ and $0.01$ respectively on the equivariant term.

**Linear evaluation**    We train a linear classifier on frozen features for 100 epochs with a batch size of 512 for CIFAR10, CIFAR100, and STL10 datasets. To optimize the classifier, we employ the Adam optimizer with a learning rate of $1e^{-3}$ and a weight decay of $1e^{-6}$. In the case of ImageNet100, we train the linear classifier for 60 epochs using a batch size of 128. We initialize the learning rate to 30.0 and apply a step scheduler with an annealing rate of 0.1 at epochs 30, 40, and 50. The remaining hyper-parameters are retained from the official code.

## G   ADDITIONAL EXPERIMENTS

### G.1   ABLATING LOSS TERMS

### G.2   HISTOGRAM FOR LOSS ABLATION.

To accompany Figure 10, this section plots the cosine similarity between positive pairs. We provide two plots for each experiment: the first plots the *histogram* of similarities of positive pairs drawn from the test set; the second plots the *average* positive cosine similarity throughout training. The results are reported in Figures 11, 12, 13, 14, 15, 16.

### G.3   ADDITIONAL PROTEIN TRAJECTORIES

In the protein cloud experiment in Section 5.1, we use the Protein Data Bank (PDB)—the single global repository of experimentally determined 3D structures of biological macromolecules and their complexes, containing approximately 130,000 samples. The core objective is to assess whether our method can effectively encode the SO(3) manifold, a highly challenging and critical aspect in drug discovery. Each input point cloud is transformed through action of the SO(3) group. Consequently, the desirect 2D trajectory is a circle corresponding to rotation along each of three orthogonal axes. This is consistent with the structure of the SO(3) manifold. Figures 17, 18, 19 and 20 illustrate

---

[*] https://github.com/facebookresearch/moco

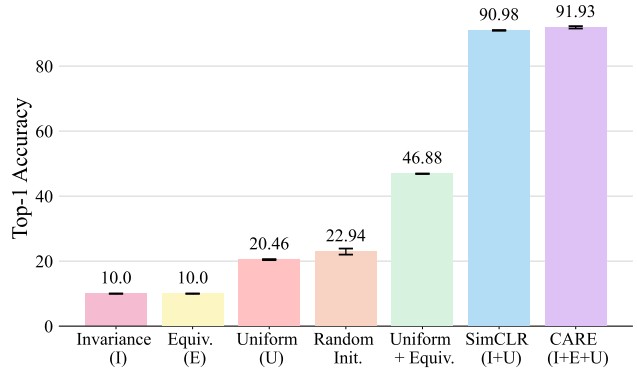

Figure 10: Ablating different loss terms. Combining $\mathcal{L}_{equi}$ with a uniformity promoting non-collapse term suffices to learn non-trivial features. However, optimal performance is achieved when encouraging *smaller* rotations, as in CARE. ResNet-50 models pretrained on CIFAR10 and evaluated with linear probes.

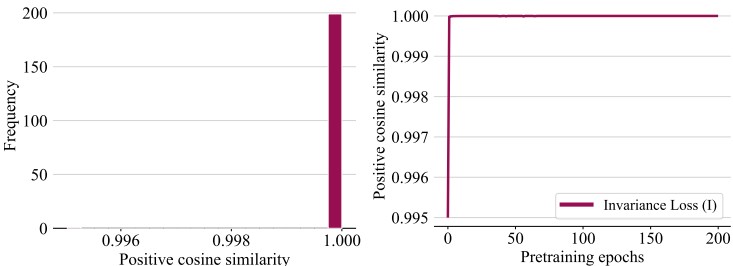

Figure 11: (left) Histogram of positive cosine similarity values at the end of pre-training using the invariance loss; (right) Evolution of positive cosine similarity values over pre-training epochs using the invariance loss

additional trajectories observed through the embedding space of a DeepSet trained with the CARE, SimCLR, $\mathcal{L}_{unif}$ and $\mathcal{L}_{inv}$ loss respectively.

### G.4 ADDITIONAL QUALITATIVE ASSESSMENT OF EQUIVARIANCE

Figure 21 qualitatively assesses sensitivity of the representation to changes in color in the Flowers-102 dataset. We compare CARE to a popular equivariant baseline, EquiMOD (Devillers & Lefort, 2023). As anticipated, both CARE and EquiMOD demonstrate sensitivity to changes in query color and hence are *equivariant* to color. However, as depicted in Figure 21, EquiMOD's representation exhibits

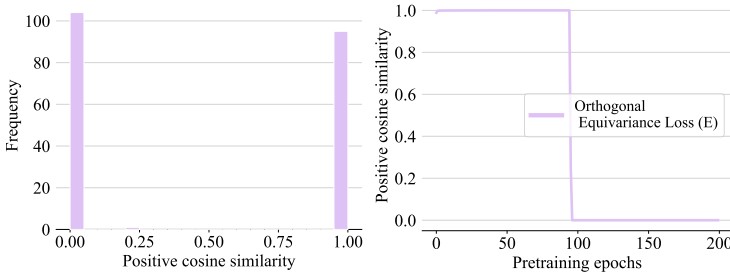

Figure 12: (left) Histogram of positive cosine similarity values at the end of pre-training using the orthogonal equivariance loss; (right) Evolution of positive cosine similarity values over pre-training epochs using the orthogonal equivariance loss

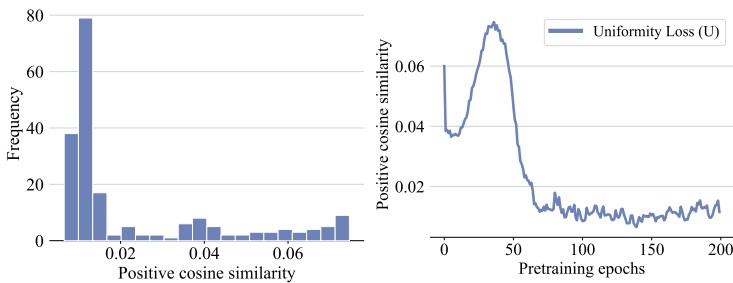

Figure 13: (left) Histogram of positive cosine similarity values at the end of pre-training using the uniformity loss; (right) Evolution of positive cosine similarity values over pre-training epochs using the uniformity loss

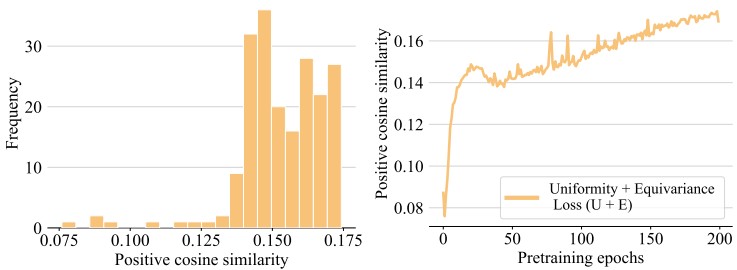

Figure 14: (left) Histogram of positive cosine similarity values at the end of pre-training using the Uniformity + Equivariance loss; (right) Evolution of positive cosine similarity values over pre-training epochs using the Uniformity + Equivariance loss

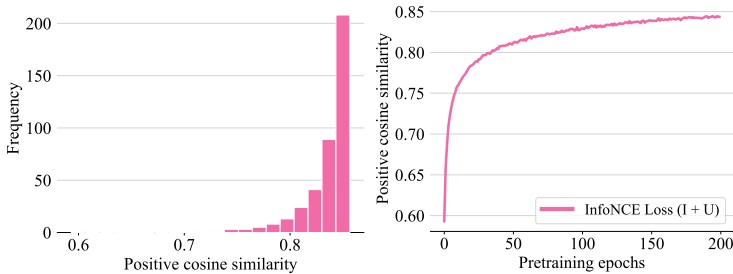

Figure 15: (left) Histogram of positive cosine similarity values at the end of pre-training using the InfoNCE (invariance + uniformity) loss; (right) Evolution of positive cosine similarity values over pre-training epochs using the InfoNCE loss

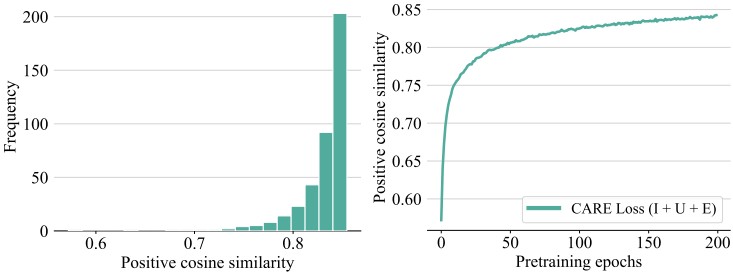

Figure 16: (left) Histogram of positive cosine similarity values at the end of pre-training using the CARE (InfoNCE + orthogonal equivariance) loss; (right) Evolution of positive cosine similarity values over pre-training epochs using the CARE loss

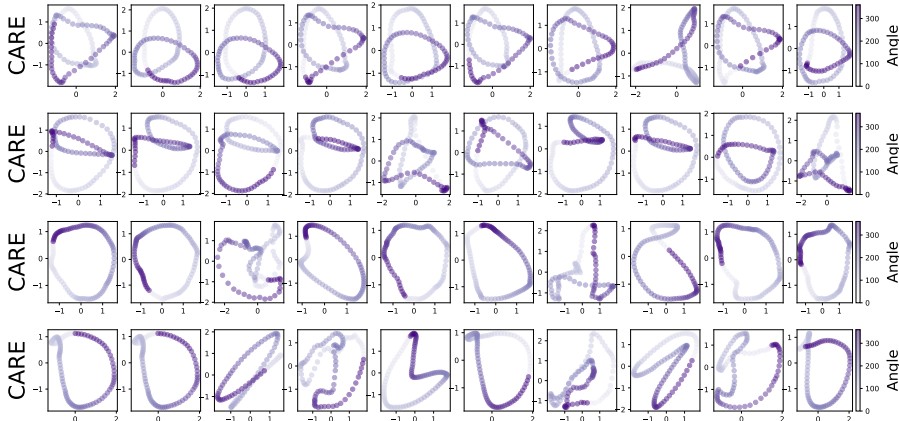

Figure 17: Additional trajectories through the embedding space of a DeepSet trained with CARE.

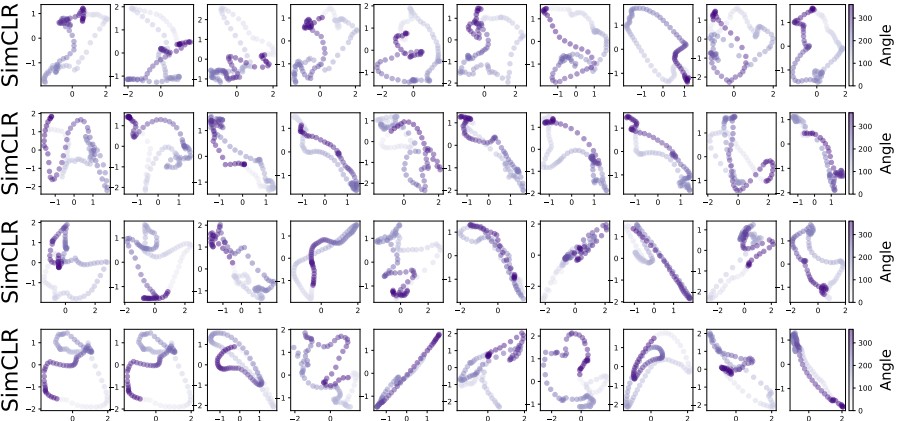

Figure 18: Additional trajectories through the embedding space of a DeepSet trained with SimCLR.

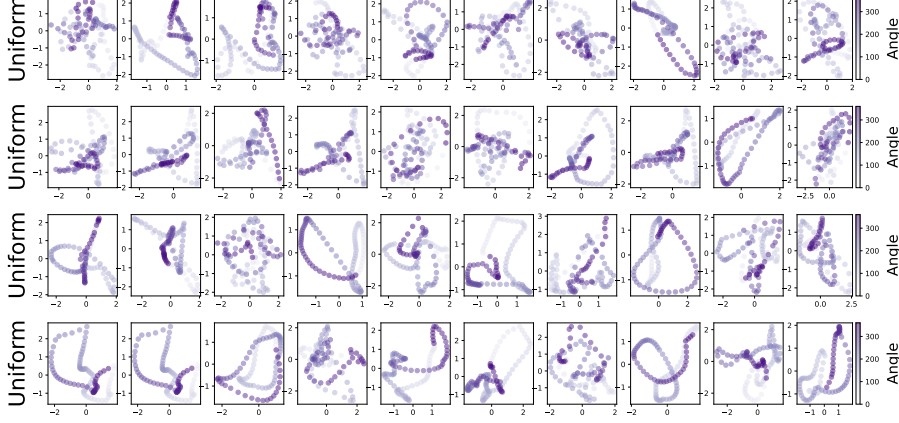

Figure 19: Additional trajectories through the embedding space of a DeepSet trained with the uniformity loss $\mathcal{L}_{\text{unif}}$.

nearest neighbors with significantly different shades (e.g., red and orange) compared to those learned by CARE, which are closer in color to the query images.

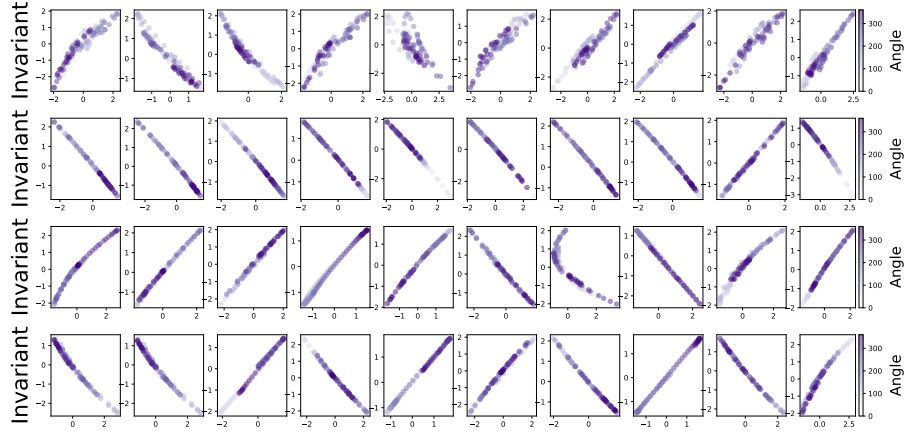

Figure 20: Additional trajectories through the embedding space of a DeepSet trained with the invariance loss $\mathcal{L}_{\text{inv}}$.

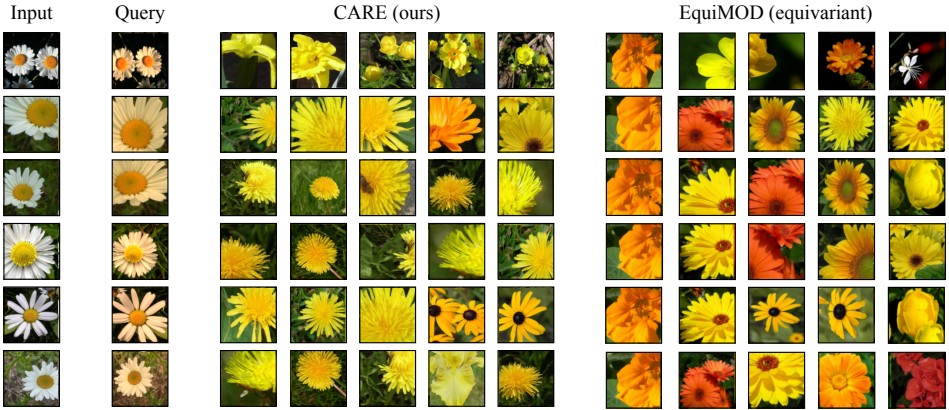

Figure 21: CARE exhibits sensitivity to features as with other equivariance-based contrastive methods (e.g., EquiMOD). For each query input, we retrieve top 5 nearest neighbors in the embedding space.

## H  ADDITIONAL DISCUSSION

**Limitations.** While our method, CARE, learns embedding spaces with many advantages over prior contrastive learning embedding spaces, there are certain limitations that we acknowledge here. First, we do not provide a means to directly identify the rotation corresponding to a specific transformation. Instead, our approach allows the recovery of the rotation by solving Wahba's problem. However, this requires solving an instance of Wahba's for each augmentation of interest. Future improvements that develop techniques for quickly and easily (i.e., without needing to solve an optimization problem) identifying specific rotations would be a valuable improvement, enhancing the steerability of our models. Second, it is worth noting that equivariant contrastive methods, including CARE, only achieve approximate equivariance. This is a fundamental challenge shared by all such methods, as it is unclear how to precisely encode exact equivariance. The question remains open as to a) whether this approximate equivariance should be considered damaging in the first place, and if so, b) whether scaling techniques can sufficiently produce reliable approximate equivariance to enable the diverse applications that equivariance promises. Addressing this challenge is a crucial area for future research and exploration in the field. Each of these limitations points to valuable directions for future work.

**Broader impact.** Through our self-supervised learning method CARE we explore foundational questions regarding the structure and nature of neural network representation spaces. Currently, our approaches are exploratory and not ready for integration into deployed systems. However, this line of work studies self-supervised learning and therefore has the potential to scale and eventually contribute to systems that do interact with humans. In such cases, it is crucial to consider the usual

safety and alignment considerations. However, beyond this, CARE, offers insights into algorithmic approaches for controlling and moderating model behavior. Specifically, CARE identifies a simple rotation of embedding space that corresponds to a change in the attribute of the data. In principle, this transformation could be used to "canonicalize" data, preventing the model from relying on certain attributes in decision-making. Additionally, controlled transformations of embeddings could be used to debias model responses and achieve desired variations in output. It is important to note that while our focus is on the core methodology, we do not explore these possibilities in this particular work.

