# OpenReview forum: "Structuring Representation Geometry with Rotationally Equivariant Contrastive Learning"
_ICLR.cc/2024/Conference — ICLR 2024 poster_

### Official Review · Reviewer_avuo · 2023-10-31

**Soundness:** 2 fair
**Presentation:** 2 fair
**Contribution:** 3 good
**Rating:** 6
**Confidence:** 4

**Summary:**

The paper introduces a contrastive learning framework called CARE, which stands for Contrastive Augmentation-induced Rotational Equivariance. CARE extends the InfoNCE loss by including a term that enforces an equivariant constraint with respect to the image transformations employed during training, as opposed to the traditional contrastive learning that attempts to induce invariance.

The central concept behind CARE is to ensure that transformations in the input space correspond to local orthogonal transformations in the representation space, a property known as orthogonal equivariance. This is achieved by minimizing a loss term (L_equi) that encourages the angles between pairs of images and their transformed counterparts to remain ideally the same, thus promoting orthogonal equivariance. The authors show theoretically that satisfying their proposed loss (i.e., L_equi = 0) implies orthogonal equivariance.

To avoid a common issue where all data points collapse to the same location in the representation space, the authors introduce an additional term (L_uni) inspired by the work of Isola and Wang. L_uni encourages representations within a batch to be evenly distributed. However, the combination of L_equi and L_uni alone underperforms compared to traditional contrastive learning frameworks like SimCLR. To address this, the authors introduce a term (L_inv) traditionally used to induce invariance with respect to the applied transformations. The final loss function comprises these three terms: \lambda * L_equi (minimize angle differences) + L_inv (minimize differences among positive pairs) + L_uni (maximize uniformity).

 The authors assess CARE on a protein point cloud dataset (Protein Data Bank), they also demonstrate that CARE can better induce orthogonal equivariance compared to SimCLR using metrics like Whaba's problem and Relative Rotational Equivariance applied to the embedding of a Resnet-18 architecture trained on CIFAR-10. They qualitatively compare CARE and SimCLR in an image retrieval task, and finally evaluate CARE against SimCLR, MoCo-V2, and BYOL in the linear probing image classification task on various datasets: CIFAR10, CIFAR100, STL10, and ImageNet100.

**Strengths:**

The paper tackles an important issue: that of training more generic representation. I appreciate the idea of mixing invariance and equivariance for this objective.

I also find the idea of representing equivariance using angle preservation as a neat idea.

**Weaknesses:**

I think this work has some interesting intuitions and hope that if it is not accepted the authors will continue to make it stronger. While I like the idea of mixing invariance and equivariance I think this work could be strengthened by a more thoughtful use of the contrastive framework and a more solid experimental evaluation. Let me elaborate on both these points. Since the experimental evaluation is, in my opinion, a more impactful weakness I start there, and follow with the use of the contrastive framework.

The experimental protocols should be improved in order to highlight the benefit of CARE. Results are not clearly showing why one should adopt CARE, some key experiments for this work are missing (transfer learning), and other experiments do not seem comparable with previously presented results.  Specifically,

- The experiments that use the Wahba’s Problem and the Relative Rotation Equivariance should contain some stronger baselines. Wouldn't the authors say that it is expected that CARE achieves better results in terms of equivariance compared to SimCLR (that does not try to achieve orthogonal equivariance)? SimCLR could be a lower bound but it is difficult to say if the results shown by CARE are "good" as no other strong comparison is provided. A stronger result would be to show that CARE is comparable or superior to other equivariant inducing algorithms.

- The results on linear probing seems to report results on a not very common setting (training on CIFAR-10, 100, STL, ImageNet100, as opposed to training on ImageNet or ImageNetTiny and testing a linear probe on those datasets). For example see table Table 3 of BYOL, or Table 8 of SimCLR , or Table 3 (RRC column) of Duet. All these results were obtained training on ImageNet and testing on other datasets (including Cifar10 and Cifar100) and report much stronger results. I believe these results (i.e. Transfer Learning) are crucial for this work because they could support the claim that by using equivariance one can learn more generic and transferable features. Training and testing on the same dataset it is less interesting in my opinion.

- Even for the few results that can be found about training SimCLR on CIFAR-10 (for example) the results reported seem below previously reported numbers. The original SimCLR paper reports results on CIFAR 10 in the Appendix and the accuracy is 94% (compared to 90.98%). My experience with contrastive frameworks is that once the pre-training pipeline is optimized and the training scales up (in terms of datasets, batch size, and epochs) the initial advantages one might observe by tweaking the model reduce until they (very often) disappear altogether. It would be great to show that this does not happen with equivariance.

- On the protein point clouds experiments I find unclear to assess what is the desired (correct) trajectory. Perhaps this is due to my lack of experience with this protein task but the manuscript could explain more clearly what is the expected result (Figure 3). Also, it is known that SimCLR requires large batches (hence large datasets), I am not familiar with this dataset but what I wonder if its size makes it a suitable application for pre-training with SimCLR. Maybe a better test would be to pre-train on a larger dataset and fine tune on this dataset?

- Similarly, the qualitative results on the image retrieval tasks are difficult to assess. I thought the results from SimCLR were better. This task should be explained more clearly and it would be even better to employ some quantitative evaluation (if possible). I also do not fully understand what is the role of the “input” compared to that of the “query”.

In terms of the contrastive learning framework, if I understand correctly all three terms of the proposed algorithm act on the same representation space (after projection head). This however introduces an ambiguity: is the objective to induce invariance or equivariance? While it is true that invariance satisfies equivariance it is also true that satisfying equivariance through invariance leads to the loss of information typical of the invariance-inducing algorithms (as the authors correctly mention in the paper). It is also why the common contrastive learning algorithms based on invariance impose this constraint after the projection head but use the representation before (where "hopefully" invariance is not achieved strongly). Therefore, while the equivariance idea is great and the use of angles is an interesting idea, its application seems to have "just" a regulation effect (especially given that the \lambda value is so small). A more interesting (albeit I recognize this comment falls in the realm of speculations) use of the contrastive framework would be to impose equivariance on the embedding before the projection head (the embedding that are actually used for downstream tasks) and invariance after the projection head. By doing so, one can hope to take full advantage of the equivariant properties while still leveraging the constructive learning framework and the benefit of the invariance loss.

**Questions:**

My question mostly revolve around the weaknesses listed above. Is there any misunderstanding in the list I provided above?

In addition
- Would you agree that it make sense to compare Wahba’s Problem and the Relative Rotation Equivariance with another equivariance inducing method (in addition to SimCLR which is not inducing equivariance)? And that without this comparison it's difficult to assess "how good" the results shown are.
- Would you be able to provide transfer learning results? Would you agree that those are ideal to show that CARE is learning more generic features?
- Could you resolve the issue on the CIFAR-10 classification results? Why SOTA report 94% accuracy while your version is at 90%?
- Could you provide more clarity on the protein folding experiment?
- Could you provide more clarity (and maybe quantitative metrics) on the image retrieval task?
- One of the claim is that CARE can handle composition of transformations but it is unclear if there is any experiment that support this claim. I understand that this can be proven theoretically, however, L_equi is never zero and its contribution if further reduced by the \lambda factor, hence in practice it remains to be seen if CARE can handle equivariance of composition of transformations. Or is it the case that the augmentations are random resize crop + rotation, hence the equivariance is achieved for the combination of these two?
- Lastly, not a question but I hope you take on the suggestion of trying to impose the equivariance constraint on the embeddings used for downstream task rather than the one after the projection head.

---

> ### Author Response · Authors · 2023-11-18
> **Response to the Reviewer avuo**
>
> We appreciate your critical and thorough feedback, which has been valuable in enhancing our work. We are glad you appreciated the technical novelty and analysis, and believe this could form the basis for a positive review. Below, we aim to answer their questions as concretely as possible. The changes have also been incorporated into the revised manuscript and are highlighted in magenta.
>
> ---
>
> > Comparison of CARE with another equivariant baseline for the Wahba’s problem
>
> Thank you for this valuable suggestion. We have included the following additional experiments in our revised manuscript (highlighted in magenta) to address your comments about our experimental protocols as concretely as possible.
>
>
> - Wahba error for a model trained using EquiMOD on CIFAR10 in Figure 5. As shown in the figure, CARE achieves the lowest error on Wahba's problem, highlighting its ability to learn an _orthogonally equivariant_ representation.
>
> - EquiMod, an equivariant contrastive learning baseline, in our protein point cloud experiment (Figures 3 and 4, as shown in the updated manuscript). Figure 3 provides a qualitative comparison of trajectories, demonstrating that EquiMod does not exhibit a rotational structure and is qualitatively closer to SimCLR than CARE. Figure 4 presents quantitative results, showing that CARE outperforms EquiMod on the principal component prediction task.
>
> - Qualitative assessment of the representation learned by EquiMOD in Figure 21 on Flowers102. Both CARE and EquiMOD, being equivalent baselines, show sensitivity to color.  However, EquiMOD's representation exhibits nearest neighbors with significantly different shades (e.g., red and orange) compared to those learned by CARE, which are closer in color to the query images. Note that this experiment assesses CARE's ability to learn equivariance and not orthogonal equivariance. Thus, any equivariant baseline would exhibit this sensitivity to input transformations (color variations in this case).
>
> - Additionally, we examine the quality of features learned by training the Barlow Twins [1] invariant baseline with our equivariant loss $\mathcal{L}\_{\text{equiv}}$. As shown below and in Table 1 of the revised manuscript, $\text{CARE}\_{\text{Barlow Twins}}$ outperforms its invariant counterpart on all three datasets, CIFAR10, STL10 and CIFAR100.
>
> Algorithm | CIFAR10 | CIFAR100 | STL10
> |----------|----------|----------|----------|
> Barlow Twins | 84.54 $\pm$ 0.02 | 55.54 $\pm$ 0.05 | 90.62 $\pm$ 0.02
> $\text{CARE}_{\text{Barlow Twins}}$ | **85.65 $\pm$ 0.05** | **56.76 $\pm$ 0.02** | **90.92 $\pm$ 0.01** |
>
> ---
>
>
>
> > Transfer learning experiments
>
> We acknowledge the reviewer's point regarding the common practice of evaluating linear probe performance for a model trained with ImageNet on other datasets like CIFAR10, CIFAR100, and STL10 when proposing invariant baselines. However, it's worth noting that existing equivariant research [1, 2] often conducts experiments involving pretraining and testing on the same datasets, with CIFAR10 and ImageNet being popular choices. In line with this convention, we test our approach on smaller benchmarks (CIFAR10, CIFAR100, STL10) and a larger dataset (ImageNet100) as a computationally efficient alternative to conducting experiments directly on ImageNet, which is beyond our resource limits.
>
> [1] Dangovski, Rumen, et al. "Equivariant contrastive learning." arXiv preprint arXiv:2111.00899 (2021).
>
> [2] Devillers, Alexandre, and Mathieu Lefort. "Equimod: An equivariance module to improve self-supervised learning." arXiv preprint arXiv:2211.01244 (2022).
>
> ---
>
>
> > Desired correct trajectory for the protein experiment. Could the size of the dataset or batch be responsible for lower performance in this experiment for SimCLR.
>
> - **About the dataset and desired trajectory.** In this experiment, we use the Protein Data Bank (PDB)—the single global repository of experimentally determined 3D structures of biological macromolecules and their complexes, containing approximately 130,000 samples. The core objective is to assess whether our method can effectively encode the SO(3) manifold, a highly challenging and critical aspect in drug discovery. Each input point cloud is transformed through action of the SO(3) group. Consequently, the desirect 2D trajectory is a circle corresponding to rotation along each of three orthogonal axes. This is consistent with the structure of the SO(3) manifold. FIgures 3, 17, 18, 19 and 20 illustrate additional trajectories observed through the embedding space, showing that CARE is adept at learning the SO(3) manifold. We had added additional clarification in the revised manuscript
>
> - **Failure of SimCLR.** The reason for low performance for SimCLR is its inability to encode any information about the input transformation i.e. rotation in 3D. As a consequence, it is unable to learn the orientation of a given protein. Note that the batch sizes and architectural choices were fixed for both CARE and SimCLR.

---

> ### Author Response · Authors · 2023-11-18
> **Response to the Reviewer avuo (continued)**
>
> > Number reported on some datasets such as CIFAR10 are below the previously reported numbers
>
> The 94\% top-1 accuracy reported in the appendix of the original SimCLR paper [1] corresponds to training the model with a batch size of 1024 and over 1000 epochs. Figure B.7 in this paper reports a performance of approximately 92\% using a batch size of 256 and 400 as the number of pre training epochs. A popularly referenced paper for CIFAR10 baseline numbers [2], reports a top-1 accuracy of 91.1\% for a batch size of 512 and 800 pre-training epochs. EquiMOD [3], another paper in the literature of equivariance in contrastive learning reports an accuracy of 90.98\% with a batch size of 512 and 800 pre-training epochs.
>
> As rightly highlighted in your review, the quality of representation learned by SimCLR and its performance on the downstream tasks are significantly influenced by the training batch size, choice of architecture, number of training epochs etc. Due to computational constraints, we experimented with a fixed batch size of 256 and 400 training epochs across experiments. Given our hyper-parameters, we believe our reported number of 90.98\% are consistent with the reported numbers in the above works.
>
> [1] Chen, Ting, et al. "A simple framework for contrastive learning of visual representations." International conference on machine learning. PMLR, 2020.
>
> [2] Chen, Xinlei, and Kaiming He. "Exploring simple siamese representation learning." Proceedings of the IEEE/CVF conference on computer vision and pattern recognition. 2021.
>
> [3] Devillers, Alexandre, and Mathieu Lefort. "Equimod: An equivariance module to improve self-supervised learning." arXiv preprint arXiv:2211.01244 (2022).
>
>
> ---
>
> > Qualitative results on the image retrieval tasks are difficult to assess. What is the role of “input” as opposed to “query”
> ​​
>
> In Figure 9, the “input” ($x$) corresponds to the original batch of images from the dataset, while “query” ($\tilde{x}$) refers to the transformed batch obtained using a randomly sampled augmentation $a$. The trained model receives $\tilde{x} = a(x)$ as input, and the top 5 nearest neighbors of its embedding i.e. $f(a(x))$ are extracted based on the Euclidean distance. The images corresponding to these top 5 nearest neighbor embeddings are displayed for both CARE and SimCLR.
>
> Ideally, a representation equivariant to color variations in the input space must change the retrieved results (top 5 neighbors) in response to a change in query color. From Figure 9, CARE qualitatively performs better than SimCLR in the visual retrieval task, as its retrieved neighbors have colors similar to that of the query. In contrast, the retrievals for SimCLR remain relatively static.
>
>
>
> ---
>
>
> > Use of equivariant loss before the projection head and invariance loss after the projection head
>
> Indeed, this is an excellent point. However, we tried this variant early in our testing, but found that it did not improve downstream performance. For instance we found that with ResNet18 on CIFAR10, we achieve the following kNN accuracies:
>
> Algorithm | Top-1 kNN accuracy|
> |----------|----------|
> Loss before projection | 86.0% |
> Loss after projection | **86.7%** |
>
>
> ---
>
>
> > On the consistency under compositions exhibited by CARE
>
> Thank you for raising this excellent question. We have revised the manuscript to include an approximate equivariance bound (Section A.1 and B.1) when the equivariant loss is small but non zero.  We briefly describe the results below.
>
> - To generalize Corollary 3 in our submission, assume that the loss is bounded (not exactly zero), so $||AA^\top - BB^\top|| < \epsilon^2$ in the 2-norm for some error $\epsilon^2 > 0$. Then, using Theorem 1 in [1], we can demonstrate that $\min_{R \in O(d)} ||A - BR|| < o(\epsilon)$, where $o(\epsilon) \to 0$ as $\epsilon \to 0$. As $\epsilon$ (and consequently the loss) approaches zero, we recover our exact equivariance result. For $\epsilon > 0$, we achieve an approximate equivariance up to $o(\epsilon)$.
>
> - We further show that CARE enjoys consistency under composition of transformations even under low equivariant loss. Mathematically, $\rho : \mathcal A \rightarrow O(d)$ given by $\rho(a) = R_a$ satisfies
> $||\rho(a' \circ a) - \rho(a')\rho(a)|| \leq o(\epsilon^2)$ for almost all $a,a' \in \mathcal{A}$, where $o(\epsilon) \to 0$ as the error $\epsilon \to 0$.
>
> Furthermore, we would like to clarify that CARE is tested within the standard contrastive learning framework, where each input transformation is already a composition of random crop, random horizontal flip, color jitter, random grayscaling, and Gaussian blurring.
>
> [1] Arias-Castro, Ery, Adel Javanmard, and Bruno Pelletier. "Perturbation bounds for procrustes, classical scaling, and trilateration, with applications to manifold learning." Journal of machine learning research 21 (2020).

---

> > ### Comment · Reviewer_avuo · 2023-11-20
> >
> > Thanks to these authors for taking into account my comments and providing more clarity to some of their experiments.
> > I would like to provide a few comments to these answers.
> >
> > > Comparison of CARE with another equivariant baseline for the Wahba’s problem
> >
> > It seems that EquiMod obtains even worse performance than SimCLR. Do the authors have any intuition about why this is the case?
> >
> > > Transfer learning experiments
> >
> > Admittedly, it is true that in other publications authors performed similar experiments, however, I believe that this is an important experiment to show that the equivariance achieved in the embedding produced by CARE leads to more general representations. Without such experiment the contribution, in my opinion, remains theoretical rather than adding a practical extra value.
> >
> > > Desired correct trajectory for the protein experiment
> >
> > Thank you for the clarification, now I understand the results better.
> >
> > > Number reported on some datasets such as CIFAR10 are below the previously reported numbers
> >
> > Thank you for brining clarity to these differences. They are as I suspected related to not performing the "correct and complete" optimization, which, in my experience often leads to wrong conclusions. I appreciate that training with large batch sizes and epochs is a demanding task. In my experience, however, many small gains that one can appreciate in a lower scale regime completely disappear in large scale settings. I cannot say that CARE would also become equivalent (or worse) than SimCLR but neither could I assume it will not. Similarly to my above point, showing the CARE actually is better than SimCLR would have strengthened the empirical results.
> >
> > > Qualitative results on the image retrieval tasks are difficult to assess. What is the role of “input” as opposed to “query”
> >
> > Thank for clarifying the difference between input and query. I still have one doubt. What is the desired outcome? Is an image that exhibit the same content as the input? Same color as query? Or both (content of input and color of query)? It seems to me by looking at Figure 5 and 21 that if the objective is to preserve content then both SimCLR and EquiMOD are better than CARE (3 images seems daisy in CARE, 19ish for SimCLR and 16ish in EquiMOD), if the objective is to preserve both then EquiMOD does a better job than CARE (3 images seems yellow daisy in CARE, 16ish in EquiMOD, only 2 in SimCLR). The only setting where CARE does better is to match the color (irrespective of the type of flower).
> >
> > > Use of equivariant loss before the projection head and invariance loss after the projection head
> >
> > Thanks for performing this experiment. Interesting to see that after the project is slightly better… I wonder if this is an indication that the wrong loss is being optimized… My reasoning is that if the loss was “perfect” then the embedding immediately before the optimization should be ideal, if we need to "backtrack" and truncate the model then maybe the loss is less then idea (admittedly, this comment is true for all contrastive framework, not just for CARE).
> >
> > > On the consistency under compositions exhibited by CARE
> >
> > Thanks for this analysis.
> >
> > After these clarifications I think this work is interesting from a theoretical perspective although I am skeptical about its practical use cases. Most of the experiments are still weak (although improved compared to the initial manuscript) since they do not compare with respect to the best state of art results, or settings. As mentioned above I expect these small gains to disappear once batch size and epochs are scaled up. The qualitative evaluation in my opinion is unfavorable to CARE. I do, however, recognize the merit of the proposal in terms of theoretical innovation, for this reason I will upgrade my recommendation.

---

> > > ### Author Response · Authors · 2023-11-22
> > > **Response to the latest comment by Reviewer avuo**
> > >
> > > We thank the reviewer for their continued discussion. We appreciate the reviewer for their constructive review and encouraging words regarding our theoretical perspective and regarding upgrading their recommendation.
> > > We hope that our clarifications below can further help with the concerns they raised.
> > >
> > > ---
> > >
> > >
> > > > Why is EquiMOD worse than SimCLR on the Wahba’s problem
> > >
> > > EquiMod fundamentally focuses on enforcing non-linear equivariance, which is different from orthogonal equivariance exhibited by CARE. It is possible that the emphasis on non-linear equivariance in EquiMod may negatively impact the linearity of its embedding space, leading to less orthogonal equivariance compared to SimCLR.
> > >
> > > ---
> > >
> > >
> > > > Additional transfer learning experiments
> > >
> > > We thank the reviewer for this suggestion. We believe that assessing the generalizability of the learned representations across a wide range of transformation parameters and datasets is indeed a very intriguing future direction and warrants thorough investigation.
> > >
> > > ---
> > >
> > >
> > > > Testing CARE in the large scale regime
> > >
> > > We concur with the reviewer that testing at large scale is indeed important. However, it's important to note two key points:
> > >
> > > - We cannot definitely predict if the gains because of CARE won’t persist in the large scale regime. In fact, we emphasis that our newly added baseline, Barlow Twins and $\text{CARE}_{\text{Barlow Twins}}$ were trained for 800 pretraining epochs. Notably, CARE consistently outperformed its invariant baseline, Barlow Twins across the three benchmark datasets, CIFAR10, CIFAR100 and STL10.
> > >
> > > - While the performance of SimCLR is known to be highly sensitive to batch size and pretraining epochs, other algorithms like MoCo-v2 [1] and Barlow Twins [2] have shown robustness to changes in batch size, performing well even with a batch size of 256.
> > >
> > > Following the reviewer’s suggestion and considering the constraints of available compute, we would be happy to include the baselines for 800 pretraining epochs in the camera-ready version (if requested). However, we emphasize that SimCLR on CIFAR10 achieves a top-1 accuracy of 91.1\% for a batch size of 512 and 800 pre-training epochs [3],  which is only 0.12% higher than our reported numbers.
> > >
> > > [1] Chen, Xinlei, et al. "Improved baselines with momentum contrastive learning." arXiv preprint arXiv:2003.04297 (2020).
> > >
> > > [2] Zbontar, Jure, et al. "Barlow twins: Self-supervised learning via redundancy reduction." International Conference on Machine Learning. PMLR, 2021.
> > >
> > > [3] Chen, Xinlei, and Kaiming He. "Exploring simple siamese representation learning." Proceedings of the IEEE/CVF conference on computer vision anALd pattern recognition. 2021.
> > >
> > > ---
> > >
> > >
> > > > What is the ideal behavior for models on the qualitative results on the image retrieval tasks?
> > >
> > > In this experiment the query changes just the yellowness of images. Accordingly, an ideal model would change its embeddings _only_ to reflect the new yellow hue, keeping other features as unchanged as possible, such as other colors and flower breeds. However the EquiMOD embeddings are “too sensitive” - they are sensitive to things they are not supposed to be. In this case, EquiMOD also returns 1) images with orange/much darker hue flowers, and 2) flower breeds that look very different from the original daisy.
> > >
> > > Preservation of “content” is measured in terms of the average top-1 accuracy. In particular,  all the three baseline quantitatively exhibit similar top-1 kNN accuracies (43%) on Flowers102 dataset.
> > >
> > > ---
> > >
> > >
> > > > Use of equivariant loss before the projection head and invariance loss after the projection head
> > >
> > > We agree that this merits further exploration, but as you also mention, we suspect that the underlying issues may be the same as what causes the original InfoNCE loss to require a projection head. One such cause is low-rank dimension collapse of the final embedding space (as explored in https://arxiv.org/abs/2110.09348).  Approaches such as DirectCLR that remove the projection head by explicitly avoiding dimension collapse effects may also work in our case.

---

> > > > ### Comment · Reviewer_avuo · 2023-11-22
> > > >
> > > > Thanks to these authors for their additional thoughts.

---

### Official Review · Reviewer_MvBZ · 2023-11-01

**Soundness:** 3 good
**Presentation:** 3 good
**Contribution:** 3 good
**Rating:** 8
**Confidence:** 3

**Summary:**

The paper introduces a self-supervised loss that demonstrably enforces a connection between latent embeddings of related complex augmentations through "simple" linear (rotational) transformations.
The loss is theoretically analyzed, showing that rotational equivariance emerges when scalar product in the embedding space are preserved under augmentations. Furthermore, the generalization of this fact to other bilinear forms and geometries is discussed.
The approach is assessed and demonstrated to enhance linear probing performance on image datasets. Additionally, it is qualitatively evaluated and shown to capture equivariance as intended by the suggested loss.

**Strengths:**

Overall the paper is well written and easy to follow.

The suggested loss simplifies existing equivariant techniques as it alleviates the need to learn the equivariance transformation, making it emerge solely as a result of an optimized loss term.

The method is theoretically analyzed. The analysis seems to be solid.  I appreciate the discussion about the possible generalizations to different geometries.

**Weaknesses:**

It seems that the main weakness of the paper is in the evaluation section.
First, I would expect to see both qualitative (figure 9)   and quantitative comparisons to methods that learn the equivariant transformation (such as Garrido et al.).
Secondly, the evaluation metric suggested in the Wahba’s Problem, seems to be another possible alternative loss to the suggested equivariance loss. Why shouldn’t it be used to optimize directly?
Lastly, it would also be interesting to compare to explicit parametrization of R_a.

Additionally, I found Figure 8 and the paragraph discussing relative rotational equivariance to be somewhat unclear. It appears that they could benefit from a revision to enhance clarity.

**Questions:**

No specific questions. I would appreciate a response with respect to the weakness stated above.

---

> ### Author Response · Authors · 2023-11-18
> **Response to the Reviewer MvBZ**
>
> We appreciate the reviewer's thorough and insightful review, along with their positive feedback regarding the strengths of our work, including our theoretical analysis and discussions on extending the approach to other groups and representation space geometries. In response to the review, we have highlighted the changes in magenta in the revised manuscript and address the questions raised below.
>
> ---
>
> > Why not optimize the Wahba’s Problem as an alternate loss to our equivariance loss.
>
> This is indeed an excellent question.  While Wahba's error, denoted as $\mathcal{W}\_f$ = $\min_{R \in SO(d)} || RF - F_a ||_{\text{Fro}}$, seems like a plausible loss function to enforce orthogonal equivariance, it faces practical challenges in real-world settings for two important reasons:
>
> - **Computational Inefficiency**: Solving the Wahba's problem involves obtaining a closed-form solution through the singular value decomposition of one of the two point clouds. Minimizing the Wahba's error in each iteration is computationally intensive, with a complexity of $O(m^2n, mn^2, n^3)$, where $m$ and $n$ are the dimensions of the involved matrices. In our context, the computational complexity is dominated by $O(bs * dim^2)$, where $bs$ is the batch size, and $dim$ is the dimensionality of the representation (e.g., 2048 for ResNet50). In contrast, our proposed loss achieves the same objective by operating on pairs, with the same complexity as standard invariant contrastive baselines.
>
> - **Noise and instability**: The closed-form solution to the Wahba's problem, obtained through singular value decomposition, can introduce instability and noise in calculations. These issues can accumulate over iterations, particularly when dealing with noisy data. In some cases, even minor data perturbations can lead to substantial changes in the decomposed components, especially in the smallest singular values and corresponding vectors. This noise becomes more pronounced when working with ill-conditioned matrices.
>
>
> ---
>
>
> > Both qualitative and quantitative comparisons to equivariant methods
>
> Thank you for this valuable suggestion. We have included the following additional experiments in our revised manuscript (in magenta)
>
> - EquiMod, an equivariant contrastive learning baseline, in our protein point cloud experiment (Figures 3 and 4, as shown in the updated manuscript). Figure 3 provides a qualitative comparison of trajectories, demonstrating that EquiMod does not exhibit a rotational structure and is qualitatively closer to SimCLR than CARE. Figure 4 presents quantitative results, showing that CARE outperforms EquiMod on the principal component prediction task.
>
> - Wahba error for a model trained using EquiMOD on CIFAR10 in Figure 5. As shown in the figure, CARE achieves the lowest error on Wahba's problem, highlighting its ability to learn an _orthogonally equivariant_ representation.
>
> - Qualitative assessment of the representation learned by EquiMOD in Figure 21 on Flowers102. Both CARE and EquiMOD, being equivalent baselines, show sensitivity to color.  However, EquiMOD's representation exhibits nearest neighbors with significantly different shades (e.g., red and orange) compared to those learned by CARE, which are closer in color to the query images. Note that this experiment assesses CARE's ability to learn equivariance and not orthogonal equivariance. Thus, any equivariant baseline would exhibit this sensitivity to input transformations (color variations in this case).
>
> - Additionally, we examine the quality of features learned by training Barlow Twins [1], an invariant baseline with our equivariant loss $\mathcal{L}\_{\text{equiv}}$. As shown below and in Table 1 of the revised manuscript, $\text{CARE}_{\text{Barlow Twins}}$ outperforms its invariant counterpart on all three datasets, CIFAR10, STL10 and CIFAR100.
>
> Algorithm | CIFAR10 | CIFAR100 | STL10
> |----------|----------|----------|----------|
> Barlow Twins | 84.54 $\pm$ 0.02 | 55.54 $\pm$ 0.05 | 90.62 $\pm$ 0.02
> $\text{CARE}_{\text{Barlow Twins}}$ | **85.65 $\pm$ 0.05** | **56.76 $\pm$ 0.02** | **90.92 $\pm$ 0.01** |
>
> [1] Zbontar, Jure, et al. "Barlow twins: Self-supervised learning via redundancy reduction." International Conference on Machine Learning. PMLR, 2021.
>
> ---
>
>
> > Comparison with explicit parametrization of $R_a$.
>
> We agree that comparing with the direct parameterization of augmentations $a$ would be an interesting avenue of research. However, one of the primary objectives of our work was to design an equivariant approach that _did not_ rely on parameterizing augmentations $a$ and instead learns to be equivariant to them directly from data pairs. We view this as a possible disadvantage since it relies on having convenient features describing a particular augmentation sample. Although not for rotational equivariance, this approach has been explored in related works such as EquiMod, which we compare to in Table 1.

---

> > ### Comment · Reviewer_MvBZ · 2023-11-22
> > **Response to rebuttal**
> >
> > I want to express my gratitude to the authors for providing a thorough response.
> >
> > Most of my concerns were addressed. I do encourage the authors to consider for the next revision, conducting a comparison versus Wahba’s Problem-based loss, to validate their claim about its instability.

---

> ### Author Response · Authors · 2023-11-18
> **Response to the Reviewer MvBZ (continued)**
>
> > Figure 8 and the paragraph discussing relative rotational equivariance
>
> We apologize for the confusion regarding Figure 8, which has been updated to address the initial duplication issue. It illustrates three key metrics from left to right: the relative equivariance metric $\gamma_f$, the rotational equivariance measure, and the invariance measure (as detailed in Section 4).
>
> Here are the two key observations from this plot:
>
> - The invariance measure, $\mathcal{L}_{\text{inv}}$, is comparable for both SimCLR and $\text{CARE}\_{\text{SimCLR}}$ and is non-zero, implying approximate invariance.
> - The rotational equivariance measure is significantly lower for $\text{CARE}\_{\text{SimCLR}}$ compared to SimCLR, indicating orthogonal equivariance in $\text{CARE}\_{\text{SimCLR}}$.
>
> These observations suggest that $\text{CARE}_{\text{SimCLR}}$ is not merely achieving lower equivariance error by collapsing into invariance, which would be a trivial solution of equivariance. We have included this clarification in the revised manuscript for better understanding.

---

### Official Review · Reviewer_4N3v · 2023-11-01

**Soundness:** 3 good
**Presentation:** 3 good
**Contribution:** 3 good
**Rating:** 6
**Confidence:** 3

**Summary:**

The paper introduces a novel approach to self-supervised learning, adding geometric structure to the embedding space such that input transformations correspond to linear transformations in the embedding space. In the context of contrastive learning, the study presents an equivariance objective, which theoretically ensures that data augmentations in the input space align with rotations in the spherical embedding space. This method, named CARE, not only enhances performance in subsequent tasks but also captures essential data variations, like color, which standard methods overlook.

**Strengths:**

* The proposed equivariant contrastive learning method that maps transformations of the input to local orthogonal transformations in the embedding space is new. The authors provide theoretical arguments and show empirical evidence for the desired structure of the embedding space. Both the method and its justification are novel and valuable.
* The analysis of the structure of the learned representation space is solid and interesting. Fig. 9 is a particularly insightful and sheds light on the merits of the proposed method.

**Weaknesses:**

* It would be beneficial to delve deeper into the influence of the choice of A—the space of transformations experienced during training—on the structure and caliber of the representations derived. An intriguing question to address is the method's capacity to generalize: Can the structures learned be effectively transferred to other classes or varied transformation parameter ranges? Exploring these nuances could further solidify the robustness and versatility of the method.
* While observing the performance metrics, one notices that the performance gap between CARE and SimCLR on CIFAR10 and STL10 is <1%.  It raises the question of the actual significance and practical implications of this difference. For a more comprehensive understanding, it would be great if Table 1 could also report the variance alongside the mean.
* The analysis is limited to ResNet networks. How well do the findings generalize to other architectures? How does the architecture affect the properties of the learned representations? How well does the proposed method generalize to other domains beyond the standard computer vision datasets?

**Questions:**

* In many practical applications input transformations may not preserve distances and angles. What does CARE learn in this case?
* Fig. 8 is confusing: no legend, all three plots are the same (?).

---

> ### Author Response · Authors · 2023-11-18
> **Response to the Reviewer 4N3v**
>
> We are grateful to the reviewer for the time they put in to review our work. We are glad to see that they recognize several strengths in our work, including theoretical guarantees, comprehensive empirical evaluation, and qualitative measures for assessing the structure of learned representation. Below, we share our thoughts on the questions asked.
>
> ---
>
>  > Can the structures learned be effectively transferred to other classes or varied transformation parameter ranges?
>
> Thank you for your valuable suggestion. We would like to emphasize that the key aim of this paper is to learn a truly equivariant representation which maps complex input transformations to isometries in the embedding space. Assessing the generalization capabilities of the learned representation across a wide range of transformation parameters is indeed a very intriguing future direction and warrants thorough investigation.
>
> ---
>
>
> > Reporting variance along with the mean due to small performance differences on CIFAR10 and STL10.
>
> This is indeed a crucial point, and we have updated the manuscript to include both the average performance and standard deviations for clarity. As shown in the results summarized below, CARE outperforms its invariance counterparts by more than the corresponding standard deviation.
>
> Algorithm | CIFAR10 | CIFAR100 |STL10 | ImageNet100 |
> |----------|----------|----------|----------|----------|
> SimCLR | 90.98 $\pm$ 0.10| 66.77 $\pm$ 0.34 | 84.19 $\pm$ 0.13| 72.79 $\pm$ 0.08
> $\text{CARE}_{\text{SimCLR}}$ | **91.92 $\pm$ 0.12** | **68.05 $\pm$ 0.28** | **84.64 $\pm$ 0.29** |**76.69 $\pm$ 0.08**
> | | | | |
> MoCo-v2 | 91.95 $\pm$ 0.05| 69.88 $\pm$ 0.23| - | 73.50 $\pm$ 0.19
> $\text{CARE}_{\text{MoCo-v2}}$| **92.19 $\pm$ 0.01** | **70.56 $\pm$ 0.15** | 88.97 $\pm$ 0.48 | **74.30 $\pm$ 0.07**
>
> ---
>
>
>
> > On the generalization to other architectures and domains beyond computer vision.
>
> ResNets are a popular and common choice of architecture for contrastive learning and are also used in our experiments.  To address the concern of the reviewer, we run additional baselines on the CIFAR10 dataset using both SimCLR and $\text{CARE}_{\text{SimCLR}}$ with DenseNet121. CARE outperforms the baseline on top-1 linear probe accuracy as shown below
>
> Algorithm | Linear Probe Accuracy
> |----------|----------|
> SimCLR | 82.45
> $\text{CARE}_{\text{SimCLR}}$ | **83.56**
>
> Further, in the original manuscript, we also demonstrate the effectiveness of our approach in learning the SO(3) manifold on 3D protein structures using a DeepSet architecture, which is significantly different from 2D vision models. Qualitative analysis to assess the learned structure in the representation is also conducted for the Flowers 102 dataset apart from the standard linear probe experiment on the vision datasets.
>
> ---
>
>
> > In many practical applications input transformations may not preserve distances and angles. What does CARE learn in this case?
>
> In our setup, we train a model to map complex input transformations like random cropping, color jitter, and random flipping to orthogonal transformations in the embedding space. Note that these input transformations typically do not preserve distances or angles for a general encoder. Even so, the underlying mechanism of CARE - to transform data into an embedding space where the complex input transformations become orthogonal transformations. This is provably possible for any transformation that belongs to a compact Lie group, so long as the embedding space dimension is large enough (this is also discussed after Corollary 1).
>
> ---
>
>
> > Fig. 8 is confusing
>
> We apologize for the confusion regarding Figure 8, which has been updated to address the initial duplication issue.

---

> > ### Comment · Reviewer_4N3v · 2023-11-22
> >
> > I would like to thank the authors for the effort they put into addressing my concerns.

---

### Official Review · Reviewer_Dtcq · 2023-11-08

**Soundness:** 4 excellent
**Presentation:** 4 excellent
**Contribution:** 3 good
**Rating:** 8
**Confidence:** 3

**Summary:**

The authors consider learning an embedding $f$ for high-dimensional data into a
structured latent space using (unsupervised) contrastive-learning-type methods.
They propose a new loss function (in implementation, effectively a new
regularizer) for learning this embedding, which builds on prior work on
equivariant regularizers. The regularizer asks not for the embedding $f$ to be
invariant to augmentations $a(x)$ of the input $x$, as in previous contrastive
learning methods such as SimCLR, but that it maps those augmentations to simple
linear transformations of the input, say $f(a(x)) = T_a f(x)$; it does this
indirectly via an equivariance regularizer that the authors prove yields the
aforementioned structure on $f$ when exactly minimized. The precise
implementation becomes using this regularizer on top of existing contrastive
learning losses, such as SimCLR or InfoNCE, to prevent trivial embeddings and to
encourage (in some sense) $a \mapsto T_a$ to be "continuous". Experiments
demonstrate on simple datasets that the method improves the embeddings with
respect to two equivariance metrics; it improves linear probe performance on
ImageNet-100 scale image classification over SimCLR/MoCo-v2 (in the best case,
by a significant margin); and that its design choices are necessary for these
improvements, via ablations.

**Strengths:**

- The paper is very well written. Conceptual explanations are clear, theoretical
  results are precisely phrased and explained (mercifully, the representation
  theory is kept to an absolute minimum, which seems uncommon in this area), and
  the general writing is engaging and compelling.

- The experimental evaluation is solid: it proposes reasonable metrics to assess
  learned equivariance, shows that CARE improves over baselines on these
  metrics, demonstrates improved linear probe performance at reasonable scale,
  and gives some useful ablations in the appendix.

- The inclusion of a theoretical (mostly conceptual-type) basis for the method
  is valuable, and completes a well-rounded presentation of the method and its
  motivations.

**Weaknesses:**

- The use of both an invariant and equivariant loss in the overall objective
  function seems conceptually strange (although the ablations show it leads to
  superior performance). I would like to understand what might be being learned
  with this combination of losses -- my reading of the explanation in section 3
  is that, among embeddings that *minimize* the equivariant regularizer (following
  proposition 1), those that achieve a small invariant loss will prefer small,
  rather than large, orthogonal transformations. However, it is not clear to me
  why this setting should arise in experiments -- why not a situation where the
  invariance loss is minimized, and the equivariant regularizer is only small?
  I am also unsure how this connects with the discussion in the "Relative
  rotational equivariance." paragraph later.


- It would be ideal if the authors could assert theoretically some degree of
  approximate invariance given approximate minimization of their regularizer
  (see a comment to this effect below). It seems important to precisely
  understand what happens in this setting, given the fact that a mixture of
  equivariant and invariant losses are required for strong practical
  performance.


### Minor issues

- After equation (4): better to not reference a figure in the appendix without
  adding something like "In the appendix, we show ..."

- The claim after Proposition 1 that "[c]onsequently, low [CARE] loss converts
  'unstructured' augmentations in input space to have a structured geometric
  interpretation as rotations in the embedding space" does not seem to follow
  from Proposition 1 or the preceding discussion (since the proposition requires
  exact minimization of the loss, rather than just a low value). It is not clear
  to me that the proof generalizes, because it relies on an external result from
  invariant theory (which might be algebraic).

- Bottom of page 5: principle -> principal

**Questions:**

- Is there an "approximate" version of the results from invariant theory that
  the authors use to establish their theoretical results? If one digs into the
  proof of the result from invariant theory (say, specialized to
  $\mathrm{SO}(n)$), what obstructions are there to having an "approximate"
  version of the result?

---

> ### Author Response · Authors · 2023-11-18
> **Response to the Reviewer Dtcq**
>
> We thank the reviewer for their thought-provoking review and encouraging words of appreciation for our work. We have endeavored to address your concerns as concretely as possible, with the changes in the revised manuscript highlighted in magenta.
>
> ---
>
>
>  > The use of both an invariant and equivariant loss in the overall objective function seems conceptually strange
>
> - As highlighted in our original submission, the core idea behind this conceptualization is based on our hypothesis that small perceptual changes in data, resulting from data augmentations, should correspond to small perturbations in embeddings. However, solely minimizing the equivariance and uniformity loss $\mathcal{L}{\text{equiv}} + \mathcal{L}{\text{unif}}$ does not guarantee this. To address this, we introduce an additional constraint by enforcing localized transformations in the representation space. A natural choice in contrastive learning is to minimize the invariance loss $\mathcal{L}_{\text{inv}}$ to push the embeddings of pairs of data closeby. In essence, our model aims to learn approximate equivariance, where augmentations in input space translate to _small_ and _orthogonal_ transformations in the representation space.
>
> - Relative rotational equivariance $\gamma_f$ is the ratio of an orthogonally equivariant measure to the invariance loss. Specifically, since invariance $f(a(x)) = f(x)$ is a trivial solution of our equivariant loss, through $\gamma_f$, we measure the degree to which $f(a(x)) = R_af(x)$ where $R_a \neq I_d$.  A lower value of $\gamma_f$ indicates a higher degree of orthogonal equivariance, signifying non-trivial equivariance.
>
>
> ---
>
>
> >  Theoretical guarantees of CARE given approximate (and not exact) minimization of the regularize
>
> We thank the reviewer for raising this excellent question. Indeed, we can provide an approximate equivariance bound when the equivariant loss is non zero. The changes are highlighted in Section A.1 and B.1 of the Appendix (in magenta) in the revised manuscript and are briefly described below.
>
> - To generalize Corollary 3 in our submission, assume that the loss is bounded (not exactly zero), so $||AA^\top - BB^\top|| < \epsilon^2$ in the 2-norm for some error $\epsilon^2 > 0$. Then, using Theorem 1 in [1], we can demonstrate that $\min_{R \in O(d)} ||A - BR|| < o(\epsilon)$, where $o(\epsilon) \to 0$ as $\epsilon \to 0$. As $\epsilon$ (and consequently the loss) approaches zero, we recover our exact equivariance result. For $\epsilon > 0$, we achieve an approximate equivariance up to $o(\epsilon)$.
>
> - We further show that CARE enjoys consistency under composition of transformations even under low equivariant loss. Mathematically, $\rho : \mathcal A \rightarrow O(d)$ given by $\rho(a) = R_a$ satisfies
> $||\rho(a' \circ a) - \rho(a')\rho(a)|| \leq o(\epsilon^2)$ for almost all $a,a' \in \mathcal{A}$, where $o(\epsilon) \to 0$ as the error $\epsilon \to 0$.
>
> [1] Arias-Castro, Ery, Adel Javanmard, and Bruno Pelletier. "Perturbation bounds for procrustes, classical scaling, and trilateration, with applications to manifold learning." Journal of Machine Learning Research 21 (2020).
>
> ---
>
>
> > Minor issues and typographical errors
>
> We thank you for your attention to detail. We have made the following corrections in the revised manuscript.
>
> - *Referencing of figures in appendix*: Thank you for pointing this out. We have fixed this in the revised manuscript.
> - *Regarding low CARE loss guarantees*: We concur with the reviewer and have made the correction in the revised manuscript.
> - *Typographical error*: We are changed ‘principle’ to ‘principal’ at the bottom of page 5

---

> > ### Comment · Reviewer_Dtcq · 2023-11-22
> > **thanks**
> >
> > Dear authors,
> >
> > Thanks for your thorough response to my review, and for engaging with it. I am happy to see that perturbation bounds were available to generalize the 'exact' minimization result that was present in the submission. I will increase my score.

---

### Author Response · Authors · 2023-11-18
**Summary of key revisions**

We thank the reviewers for their time and expertise in evaluating our paper. Their perceptive remarks and constructive feedback have been valuable in improving our work. In response, we have made several key revisions to address their concerns and enhance the support for our claims. Below is a brief summary.

* Additional benchmarking against an equivariant baseline, providing both quantitative and qualitative assessments (Section 5.1, 5.2 and 5.3)
* Addition of an invariant baseline, Barlow Twins and its equivariant counterpart trained using CARE, called $\text{CARE}_{\text{Barlow Twins}}$, for linear probe classification (Section 5.4)
* Theoretical guarantees of CARE under approximate minimization of the equivariance loss.
* Clarification of key aspects of our algorithm, such as the presence of both invariance and equivariance loss, as well as clarifications in assessments, such as the protein point cloud experiment, Wahba’s problem, etc.

---

### Meta-Review · Area_Chair_Parw · 2023-12-04

**Metareview:**

The paper proposed a new CARE contrastive learning approach. CARE extends the InfoNCE loss by including an additional term to enforce an equivariant constraint w.r.t. the image transformations, i.e., transformations in the input space correspond to local orthogonal transformations in the representation space.  CARE is compared with SOTA on the image retrieval datasets including CIFAR10, CIFAR100, STL10, and ImageNet100.

This paper received initial scores of 8, 6, 6, 5. The negative reviewer discussed lots of issues and weaknesses of the paper, but the authors have done a good job in the rebuttal phase. Reviewers are generally satisfied with the authors' response, and the scores are raised to 8, 8, 6, 6 for a clear acceptance.

Please remember to fix all issues pointed out by the reviewers when preparing the final version.

**Justification For Why Not Higher Score:**

n/a

**Justification For Why Not Lower Score:**

n/a

---

### Decision · Program_Chairs · 2024-01-16

Accept (poster)